# Comprehensive Metabolome and Transcriptome Analysis of *Populus davidiana* and Its Response to Drought Stress

**DOI:** 10.3390/biology14111574

**Published:** 2025-11-10

**Authors:** Yanmin Wang, Zhihui Yin, Haixia Li, Jing Li, Chengbo Guo, Zhenghua Li, Haifeng Zhang, Hongmei Wang, Hui Bai

**Affiliations:** 1Forestry Research Academy of Heilongjiang Province, Harbin 150081, China; wangyanmin1919@163.com; 2Key Laboratory of Fast-Growing Tree Cultivating of Heilongjiang Province, Forestry Research Institute of Heilongjiang Province, Harbin 150081, China; 3State Key Laboratory of Tree Genetics and Breeding, Northeast Forestry University, Harbin 150040, China

**Keywords:** *Populus davidiana*, drought stress, transcriptome, metabolome, transcriptional regulation, reactive oxygen species

## Abstract

Drought stress severely limits the growth and survival of poplars. The physiology, transcriptome, and metabolome of poplar trees under drought stress were thoroughly examined in this work. Major impacts on poplar physiological functions were identified, such as a decrease in chlorophyll content and an imbalance in ROS clearance. A large number of DEGs were enriched in photosynthesis, chlorophyll decomposition and synthesis, as well as secondary metabolism-related pathways. These findings provide a basis for developing *P. davidiana* cultivars that can withstand drought and thrive in dry environments.

## 1. Introduction

As the world’s biggest terrestrial ecosystem, forests are essential to the carbon cycle. They are the resources and environment that humans rely on for survival and development [1]. However, the survival and growth of trees are limited by various adverse conditions. Drought is one of the primary elements that significantly threaten forest yields, and severe drought can even lead to the death of trees [2]. Under drought conditions, the amount of water available for trees to absorb decreases, photosynthesis is disrupted, and biomass decreases [3]. Moreover, as trees absorb water through their roots and transport it to the aboveground parts through their xylem, insufficient water can easily cause hollowing and blockage of the xylem [4]. Brodribb found that water columns were ruptured under tension in leaf veins, cutting off the water supply to local leaf tissues. This immediately led to acute cell dehydration and irreversible damage, resulting in forest death [5]. At the same time, drought causes an imbalance in the mechanism for removing reactive oxygen species (ROS), leading to lipid peroxidation in membranes and cell damage [6].

During the long-term evolution process, trees have developed resistance mechanisms to cope with drought stress, such as closing stomata, reducing transpiration, and minimizing water loss [7]. Concurrently, drought signals are transmitted through the signal transduction network to induce the production of certain genes and encourage the synthesis of proline, sucrose, and osmoregulatory proteins, while maintaining cell water under water-deficient conditions [8]. On the other hand, plants reduce membrane damage through lipid remodeling and enhance the antioxidant capacity to eliminate excessive ROS in plants, protecting them from oxidative stress caused by drought stress [9]. Plant antioxidants mainly include antioxidant enzymes such as catalase (CAT), peroxidase (POD), superoxide dismutase (SOD), and ascorbate peroxidase (APX), as well as non-enzymatic antioxidants such as glutathione, ascorbic acid, and carotenoids, tocopherols (vitamin E), flavonoids, phenol [10]. It is generally believed that enzyme antioxidants are the initial line of defense against oxidative stress, whereas non-enzyme antioxidants constitute the second line of defense [11,12,13]. Severe oxidative damage can inactivate antioxidant enzymes and promote the synthesis of some non-enzymatic antioxidants [14].

Flavonoids represent a significant class of non-enzymatic antioxidants that not only directly clear ROS but also inhibit their production by activating other antioxidant defense responses [15]. Flavonoids participate in stress responses in significant ways. They protect against drought, temperature, nitrogen and phosphorus deficiency, ultraviolet radiation, and heavy metals [16]. Flavonoids, abscisic acid, and lignin act synergistically to inhibit stem elongation, promote the allocation of carbon assimilates to the ear, and regulate drought tolerance in maize [17]. The levels of gene expression involved in flavonoid production in the drought-excessive insensitivity (doi57) mutant were found to be significantly altered compared with a control group [18]. Consistently, the flavonoid content increased and ROS accumulation decreased. Research shows that in wheat under drought stress, the expression levels of genes involved in flavonoid production and the accumulation of flavonoids rose dramatically, with variations among types [19]. This evidence indicates that regulating the content of flavonoids can improve plant drought resistance. However, under long-term or severe stress, there is a conflict in resource allocation between growth and the stress response, which may trigger metabolic reprogramming and reduce the synthesis of high-energy-consuming secondary metabolites such as flavonoids [20,21].

In earlier research, various genes have been shown to modulate drought stress in plant responses by modulating genes related to the flavonoid synthesis pathway and altering flavonoid content [22]. In the Arabidopsis *PFG3* mutant, flavonoid synthesis and accumulation are impaired, increasing its sensitivity to drought stress [23]. In apples, MdWRKY50 promotes *MdCHS* and enhances drought stress tolerance by phosphorylating MdWRKY17 dimerization through a cascade reaction with MEK2-MPK6 [24]. *FlbZIP12* increases resistance to drought by controlling genes involved in flavonoid production in *Fagopyrum leptopodium* [25]. In poplar trees, PuC3H35 directly binds to the promoters of the anthocyanin biosynthesis gene *PuANR* and the lignin biosynthesis-related gene *PuEARLI1*. Upregulating the expression of *PuANR* and *PuEARLI1* enhances the drought resistance of *Populus ussuriensis* roots [26]. However, the upstream regulatory mechanisms of the flavonoid synthesis pathway are not fully understood, and a large number of potential regulatory genes urgently need to be identified.

Poplar trees have a significant impact on the ecology due to their strong adaptability, rapid growth, and wide use, and they are also an important plant for reforestation and for China’s Three-North Shelterbelt Initiative [27]. In 1986, it was demonstrated that exogenous genes can be expressed in poplars via *Agrobacterium tumefaciens* [28]. In 2006, the entire genome of poplar was sequenced in the genome sequencing project, making it a model species for woody plant research [29]. *P. davidiana* is widely dispersed in China, and as it is resistant to cold, drought, and barren soil, it is an important species for afforestation in mountainous areas. There is also an abundance of high-quality genome information [30]. Therefore, *P. davidiana* is an excellent material for researching how poplar responds to drought stress [31]. Given the increasing frequency and intensity of drought events driven by climate change, understanding the molecular basis of drought tolerance in *P. davidiana* is of paramount importance. This study employs an integrated approach, combining artificial drought simulation with transcriptomic and metabolomic analyses to elucidate the dynamic responses of *P. davidiana* to a progressive water deficit. We aim to (1) characterize the temporal changes in gene expression and metabolic profiles under drought stress; (2) decipher the regulatory mechanisms of key pathways, particularly flavonoid biosynthesis and lipid metabolism; and (3) identify the candidate genes and transcription factors (TFs) underlying drought adaptation. Our findings are expected to provide valuable genetic resources and a theoretical foundation for breeding drought-resistant poplar varieties through molecular-assisted strategies.

## 2. Materials and Methods

### 2.1. Plant Materials and Treatments

*P. davidiana* seedlings were grown on half-strength Murashige–Skoog (1/2 MS) media in a phytotron under conditions of 25 °C and a 16/8 h light/dark photoperiod. Poplar plants were planted in a nutrient substrate (loamy soil/vermiculite/perlite = 5:3:2) and grown up to 8 cm before being transferred to a greenhouse at a temperature of 25 °C with a 16/8 h light/dark cycle and 70–75% relative humidity. The diameter of the plastic pots was 13 cm. Throughout the entire growth cycle, plants received normal amounts of water. Specifically, when the surface of the basin soil was dry, the basin method was used to pour water until complete saturation was achieved, with any excess water drained. Two months later, potted plants were subjected to complete watering cessation treatments for 2, 4, 6, 8, and 10 d, with normally watered plants as the control. At the same time, the leaves of the seedlings were cut with sterilized scissors, placed into liquid nitrogen, and kept at −80 °C before being collected for transcriptome and metabolome analysis, as well as subsequent experimental analysis. For each treatment, three biological replicates were carried out.

### 2.2. Measurement of Physiological Indicators

Using physiological indicator test kits, H_2_O_2_ and POD were determined in accordance with the instructions of the manufacturer (Suzhou Grace Biotechnology Co., Ltd., Suzhou, China, Item number: G0168W; G0107W). The procedure outlined in the reference study was used to measure the amount of chlorophyll [32]. In short, leaf samples were extracted using a 1:1 acetone–DMSO mixture at room temperature. The absorbance was measured at wavelengths of 666 nm and 648 nm using a spectrophotometer (TU-1900, Beijing, China). Concentrations of chlorophylls were calculated using the specific equations for the DMSO solvent published in the referenced study.

### 2.3. RNA Extraction and Real-Time Quantitative PCR

Plant material RNA was extracted according to the instructions provided by the manufacturer (BioTeke, Beijing, China). The quality of the RNA extraction was assessed using 1% agarose gel and evaluated using the Bioanalyzer2100 system (Agilent Technologies, Santa Clara, CA, USA), the RNA Integrity Number (RIN) can be found in Appendix A. RNA was reverse-transcribed into cDNA using a reverse transcription kit (TransGen, Beijing, China) and then diluted 10-fold for qRT-PCR. 18S Genes are used as reference genes, and the 2^−ΔΔct^ technique was used to calculate the expression level [33]. Primer information is included in Appendix A.

### 2.4. Transcriptome Analysis

The cDNA libraries were sequenced on the Illumina sequencing platform by Metware Biotechnology Co., Ltd. (Wuhan, China). In short, mRNA was extracted from total RNA, then broken into short fragments and used as templates for first-strand cDNA synthesis. Following this, dNTPs and DNA polymerase I were incorporated to generate the second-strand cDNA, which was then purified using DNA purification magnetic beads. The final cDNA library was then obtained by PCR enrichment after fragment size selection was performed with DNA purification magnetic beads. Illumina (Illumina, Inc., San Diego, CA, USA) was used to sequence qualified cDNA libraries. fastp (V0.23.2) was used to remove adapters and non-conforming reads from raw data [34]. HISAT2 (V2.2.1) was used to align the clean data to the genome [35] (downloaded from https://www.ncbi.nlm.nih.gov/genome/13203?genome_asse-mbly_id=516658, accessed on 22 September 2023). The featureCounts (V2.0.3) [36] were used to assess the number of counts for gene alignment and calculate FPKM. DESeq2 (V1.22.1) was used to perform differential gene analysis between groups [37], and |log_2_(FoldChange)| ≥ 1 with FDR < 0.05 was established as the screening criterion for differentially expressed genes (DEGs).

### 2.5. Metabolome Analysis

Poplar leaves were used at various drought times to measure metabolite levels. After vacuum freeze-drying, the samples were ground into a homogeneous powder using a grinder (MM 400, Retsch, Haan, Germany) at a frequency of 30 Hz for 1.5 min. Subsequently, 50 mg of the powdered sample was precisely weighed using an electronic balance (MS105DΜ, Mettler Toledo, Greifensee, Switzerland) and mixed with 1200 μL of a pre-cooled (−20 °C) 70% methanol aqueous internal standard extraction solution. (For samples weighing less than 50 mg, the volume of extraction solution was added proportionally, i.e., 1200 μL per 50 mg of sample.) The mixture was vortexed six times every 30 min for 30 s. After centrifugation at 12,000 rpm for 3 min, the supernatant was collected, filtered through a microfilter with a pore size of 0.22 μm, and stored in injection vials for UPLC-MS/MS analysis. The data acquisition instrument system primarily consists of Ultra-High-Performance Liquid Chromatography (UPLC) (Milford, MA, USA) and Tandem Mass Spectrometry (MS/MS). The conditions for liquid chromatography are as follows: (1) Column: Agilent (Agilent Technologies, Inc., Santa Clara, CA, USA) SB-C18 1.8 µm, 2.1 mm × 100 mm; (2) Mobile phase: Phase A is ultrapure water (supplemented with 0.1% formic acid) and Phase B is acetonitrile (supplemented with 0.1% formic acid); (3) Elution gradient: 5% mobile phase B at 0.00 min, which linearly increased to 95% over 9.00 min before being held at 95% for 1 min; 5% mobile phase B at 10.00–11.10 min, which was then equilibrated to 5% by 14 min; (4) Flow rate: 0.35 mL/min, column temperature: 40 °C, and injection volume: 2 μL. Mass spectrometry conditions primarily include the following conditions: Electrospray ionization (ESI) source temperature: 550 °C; Ion source voltage (IS): 5500 V (positive ion mode)/−4500 V (negative ion mode); Gas I (GSI), Gas II (GSII), and Curtain Gas (CUR) set to 50, 60, and 25 psi, respectively; and a high Collision-Induced Ionization (CI) parameter. QQQ scanning employed the MRM mode with collision gas (nitrogen) set to medium. Further optimization of the declustering potential (DP) and collision energy (CE) was performed for each MRM ion pair. A specific set of MRM ion pairs was monitored per period based on metabolites eluted within that timeframe. Based on the self-built MWDB (metware database), qualitative substance analysis was carried out based on secondary spectral information. After obtaining the raw data and filling in the missing data using the k-nearest neighbors (KNN) method, the coefficient of variation (CV) peak area values were determined for the QC samples, keeping compounds with CV values below 0.5. By combining univariate statistical analysis (hypothesis testing and fold change) with the statistical analysis of many variables (PCA, principal component analysis; OPLS-DA, orthogonal partial least squares discriminant analysis), and analyzing data characteristics from multiple angles, differential metabolites can be accurately identified. Based on the OPLS-DA model (biological replicates ≥ 3), the Variable Importance in Projection (VIP) can be used to preliminarily screen metabolites with different varieties or tissues (VIP > 1). At the same time, the differential accumulation of metabolites (DAMs) can be further screened by combining univariate analysis with a fold change ≥ 2 and fold change ≤ 0.5.

### 2.6. Construction of Multi-Level Gene Regulatory Network

Based on transcriptome information, a multi-level gene regulatory network was constructed for lipid metabolism and the flavonoid synthesis pathway using the method described in the referenced study [38]. In brief, the BWERF (Bottom-to-Up Weighted Random Forest) algorithm employs a recursive random forest model to iteratively calculate the importance values of all TF regulating genes within a pathway. During each iteration, a subset of the least important TFs is excluded, while values of importance for the remaining TFs are updated and re-ranked. This process continues until only one TF remains on the list. The importance values of TFs across all pathway genes are then aggregated and fitted into a Gaussian Mixture Model to identify TFs retained in the regulatory layer immediately above the pathway layer. The TFs identified in this secondary layer are subsequently set as the new base layer to infer the next upper layer. This procedure is repeated iteratively until a multi-layer hierarchical gene regulatory network (ML-hGRN) is constructed with the desired number of layers. In this study, the recursive random forest algorithm was used, taking the expression levels of genes related to the flavonoid/lipid metabolism pathways as the input in order to infer their upstream regulatory TFs. Associations with metabolites were screened based on correlation coefficients, and gene regulatory networks (GRNs) were visualized using Cytoscape (V3.7.2) [39].

### 2.7. Statistical Analysis

The Statistical Package for the Social Sciences (VSPSS 18.0) was used to conduct statistical analysis, and one-way ANOVA was used to assess the data. A *p*-value of less than 0.05 was considered statistically significant.

## 3. Results

### 3.1. Physiological Changes Under Drought Stress in Poplar Trees

In this research, drought time was assessed on the 0, 2, 4, 6, 8, and 10 d of the experiment, respectively. The findings demonstrated that as drought stress increased, the phenotype of the seedlings changed significantly. At 2 d of drought stress, the extent of leaf withering was not immediately apparent, and at 6 d of drought stress, the degree of leaf wilting significantly increased, and the leaves began to lose their green color. On the tenth day of drought stress, the degree of leaf wilting was the strongest, with the leaves experiencing complete desiccation and chlorosis (Figure 1A). The chlorophyll content exhibited a gradual decrease under drought stress with an increase in stress time, and at 10 d of stress, the chlorophyll content was the lowest (Figure 1B). This indicates that poplar seedlings experience a physiological process from mild stress to severe damage under continuous drought stress, and their photosynthetic system may have been seriously threatened.

Cell membranes can be harmed by oxidative stress caused by drought. H_2_O_2_ is the most stable ROS, and relative conductivity is one of the most important physiological indicators for measuring the degree of cell membrane damage [40]. The H_2_O_2_ content of each poplar was roughly the same under normal growth conditions and various treatments. By contrast, the content of H_2_O_2_ gradually increased with the prolongation of stress, reaching its highest level on the tenth day (Figure 1C). The relative conductivity showed a significant increase starting from 4 d of drought stress (Figure 1D).

POD is an important antioxidant enzyme for clearing H_2_O_2_. The results of the POD activity measurement showed an increase at varying degrees after stress, reaching its highest level at 4 d (Figure 1E). This evidence indicates that drought stress induces oxidative stress in poplars, and its early-activated POD defense system collapses with continuous stress, resulting in the accumulation of H_2_O_2_ and ultimately causing irreversible oxidative damage to the cell membrane.

### 3.2. Screening of DEGs Under Drought Stress in Poplar Trees

To gain a deeper understanding of the transcriptional changes under drought stress, transcriptome analysis was performed on poplar leaves at various periods of drought stress. The 45,938,824–52,582,840 CleanReads were excluded after removing low-quality connectors, with Q20 greater than 97.87%, Q30 greater than 93.55%, and the GC content distributed between 44.04 and 45.32%. The results of the genome map show that all CleanReads maps were mapped to over 87% of the genome (Appendix A). To verify the accuracy of Solexa sequencing, we conducted qRT-PCR on five DEGs previously identified by Solexa sequencing with respective primers. These five genes were randomly selected, and RNA-seq results were consistent with qRT-PCR analysis for the five DEG profiles (Appendix A). A positive correlation coefficient was identified between qPCR and RNA-seq data (Appendix A), demonstrating its reliability. This evidence indicates the acquisition of high-quality sequencing data to support additional research.

PCA analysis shows the separation trend between samples, representing their differences. The results of 18 samples showed that with the extension of processing time, the differences between groups became more significant (Figure 2A). A total of 2127, 5334, 8894, 11,279, and 11,778 DEGs were screened out after 2, 4, 6, 8, and 10 d of drought stress, respectively. Among them, 1201, 2365, 3977, 4698, and 4876 were upregulated, while 926, 2969, 4917, 6581, and 6902 were downregulated. Except for 2 d, all downregulated genes were higher than upregulated genes (Figure 2B). This suggests that in the middle and late stages of drought stress, the transcriptional response of poplars may shift from active stress regulation to a wide range of gene expression repression to save energy and resources and adapt to long-term stress. Further exploration of gene expression differences under drought stress for various durations was conducted through Venn diagrams, and the results showed that 1270 genes were involved in drought stress, with a maximum of 2015 differentially expressed genes at 10 d (Figure 2C).

During drought stress, 1270 genes showed sustained differential expression, suggesting the potential for these genes to continue functioning in response to drought stress in poplars. Therefore, enrichment analysis was conducted on 1270 DEGs, demonstrating that DEGs showed a significant enrichment in ADP binding and terms related to the defensive reaction to fungus, the biosynthesis of flavonoids, the cellular response to hypoxia, and the flavonoid metabolic process (Figure 2D). According to KEGG enrichment analysis, DEGs were more abundant in the pathways of flavonoid biosynthesis, flavone and flavonol production, phenylalanine, tyrosine, and tryptophan biosynthesis, tropane, piperidine, and pyridine alkaloid biosynthesis. The process by which photosynthetic organisms fix carbon, the metabolism of fatty acids, unsaturated fatty acid production, and wax, cutin, and suberine biosynthesis were all assessed. The biosynthesis of amino acids and the metabolism of fructose and mannose are shown in Figure 2E.

### 3.3. Enrichment Analysis of DEGs Under Drought Stress

GO enrichment analysis was conducted to further understand the functions of DEGs under different drought stress conditions. The findings indicate that after 2 d of drought, DEGs were significantly enriched in the cellular response to hypoxia; the flavonoid biosynthetic, terpenoid metabolic, phenylpropanoid metabolic, terpenoid biosynthetic, isoprenoid metabolic and benzene-containing compound metabolic processes; the cellular response to ethylene stimulus; and the ethylene-activated signaling pathway (Figure 3A). Following 4 d of drought stress, the DEGs showed a high level of enrichment in microtubule-based movement, positive regulation of cell death, flavonoid biosynthesis, control of the metabolic process of salicylic acid, and positive regulation of response to biotic stimuli (Figure 3B). The DEGs were significantly enriched in photosynthesis, response to reduced oxygen levels, sulfur compound transport, the pigment metabolic process, sulfate transport, chlorophyll biosynthesis, photosystem II repair, photosynthesis, light harvesting in photosystem I, porphyrin-containing compound biosynthesis, and microtubule-based movement terms following 6 d of drought stress (Figure 3C). The following processes were significantly enhanced following 8 d of drought stress: photosynthesis, microtubule-based movement, the chlorophyll metabolic process, chlorophyll biosynthesis, cellular response to hypoxia, cellular polysaccharide biosynthesis, monosaccharide metabolic process, porphyrin-containing compound metabolic process, cellular response to oxygen levels, and polysaccharide biosynthesis (Figure 3D). Microtubule-based movement, cellular carbohydrate biosynthesis, cell wall biogenesis, flavonoid biosynthesis, positive regulation of response to external stimulus, xylan biosynthesis, pigment biosynthesis, photosynthetic electron transport in photosystem I, plant-type cell wall biogenesis, and cellular response to hypoxia following 10 d of drought stress were among the processes in which the DEGs were significantly enriched (Figure 3E). These changes from initial secondary metabolism (flavonoids and terpenoids) to later primary metabolism (carbohydrates and polysaccharides) showed that the metabolic center of gravity shifted systematically with stress time. Hormone signaling pathways such as ethylene and salicylic acid were activated at different stages, indicating that plants coordinate stress responses by integrating multiple signaling networks.

Drought stress can have a significant impact on certain life activities of plants. A KEGG study of enriched DEG pathways is crucial to learn more about how drought stress affects plants. The findings demonstrate that after two days of drought stress, the DEGs were primarily enriched in plant–pathogen interactions, flavonoid biosynthesis, biosynthesis of flavones and flavonols, biosynthesis of tropane, piperidine, and pyridine alkaloids. The production of phenylalanine, tyrosine, and tryptophan was observed with glycan breakdown, biosynthesis of diterpenoid, biosynthesis of monoterpenoid, metabolism of glutathione, and metabolism of fatty acids. Pathways for the production of cutin, suberine, wax, and phenylpropanoid are also shown in Figure 4A. After 4 d of drought stress, DEGs demonstrated high enrichment in the MAPK signaling pathway–plant, plant–pathogen relationship, biosynthesis of flavonoids, and biosynthesis of flavones and flavonols, In photosynthetic organisms, carbon fixation further enhanced glycan breakdown, the transduction of plant hormone signals, metabolism of ascorbate and aldarate, metabolism of starch and sucrose, and metabolism of linoleic acid, the biosynthesis of cutin, suberine, and wax, and the process of photosynthesis gluconeogenesis and glycolysis (Figure 4B). Additionally, after 6 d of drought stress, enriched pathways included plant–pathogen interactions, MAPK signaling pathway–plant relationships, plant hormone signal transduction, photosynthesis antenna proteins, starch and sucrose metabolism, porphyrin metabolism, the photosynthesis pathway, and flavone and flavonol biosynthesis. Flavonoid biosynthesis and carbon metabolism were both significantly enriched (Figure 4C). After 8 d of drought stress, the differentially expressed genes (DEGs) were primarily distributed in pathways including plant–pathogen interactions, MAPK signaling for pathway–plant, plant hormone signal transduction, photosynthesis pathway, starch and sucrose metabolism, fructose and mannose metabolism, amino sugar and nucleotide sugar metabolism, carbon metabolism, glyoxylate and dicarboxylate metabolism, and flavone and flavonol biosynthesis (Figure 4D). Following 10 d of drought stress, the differentially expressed genes (DEGs) showed significant enrichment in plant–pathogen interactions, MAPK signaling for pathway–plant, plant hormone signal transduction, photosynthesis pathway, starch and sucrose metabolism, flavone and flavonol biosynthesis, and flavonoid biosynthesis (Figure 4E). The KEGG analysis results showed that in the early stages of drought stress in poplars, the activation of defensive metabolism is the main response, followed by the shift in enrichment pathways towards photosynthesis regulation and energy metabolism reprogramming, reflecting a strategic transition from “defense” to “maintenance”. In severe drought stages, the synergistic enrichment of plant hormone signal transduction and sustained stress resistance pathways suggests that a fine regulation stage has been entered.

### 3.4. Statistics of Differentially Expressed TFs After Drought Stress

TFs are the hubs of transcriptional regulation and have important roles in various life processes [41]. After drought stress, a total of 1326 TFs were differentially expressed. The ERF family has the most TFs, with a total of 108, followed by the MYB family with 99. Following this is NAC, with 90 members (Figure 5A). Venn plot analysis showed that 140 TFs were differentially expressed at all times, while 11, 25, 24, 57, and 184 differentially expressed TFs were observed at 2, 4, 6, 8, and 10 d, respectively (Figure 5B).

Furthermore, the expression patterns of the top eight TF families were analyzed; the majority of ERF TFs were found to be upregulated, and the expression levels increased with prolonged drought time (Figure 5C). The expressions of MYB, bHLH, C2H2, and GRAS are similar. Some TFs exhibit continuous upregulation under drought stress conditions, with their expression levels continuing to rise; in contrast, others display an initial marked increase in expression levels followed by a subsequent decrease (Figure 5D–G). The expression patterns of NAC and WRKY family members are similar. After 2 d of drought stress, the expression levels decreased, followed by a rise in some members’ expression levels, but some TF expression levels remained decreased (Figure 5H,I). It is worth noting that several members of the bZIP family have significantly increased expression levels, while others have significantly decreased expression levels compared to other bZIP members (Figure 5J). These results suggest the functional diversity of TFs, which deserves further analysis.

### 3.5. Metabolome Analysis Under Drought Stress in Poplars

For a more in-depth exploration of how drought stress impacts the metabolic levels of poplar trees, changes in the metabolic levels of poplars were detected using LC-MS. The findings indicated that 131, 378, 334, 365, and 646 metabolites differentially accumulated after drought stress for 2, 4, 6, 8, and 10 d, respectively. More downregulated DAMs were observed on days 2 and 4 than upregulated ones, and as stress duration increased, more upregulated than downregulated DAMs were found (Figure 6A). To further explore the similarities and differences in the metabolic products of poplars under different durations of drought stress, analysis was performed using Venn diagrams. The findings indicated that there were 32, 62, 25, 38, and 277 DAMs at 2, 4, 6, 8, and 10 d, respectively. It is worth noting that 40 DAMs continued to accumulate differentially during drought stress (Figure 6B). Among the 40 DAMs, the highest number of flavonoids was 19, s followed by amino acids and derivatives with a count of 7, and alkaloids with a count of 4 (Figure 6C). Heatmap analysis revealed that, compared to the control, the levels of seven amino acids and derivatives and 19 flavonoids decreased during drought stress. After six days of drought stress, two alkaloids increased, one declined, and the other decreased after two and four days (Figure 6D). Overall, the levels of seven amino acids and derivatives and 19 flavonoids decreased during the stress period, indicating that prolonged drought may have inhibited the synthesis of these metabolites or promoted their consumption.

The levels of accumulated metabolites were analyzed using K-means clustering for metabolites with similar levels. A total of 1099 metabolites were divided into eight clusters (Figure 6E). It is worth noting the DAMs of clusters 3, 5, and 6. The specificity of the third cluster increased during drought stress on the tenth day, and further statistical analysis showed that there were more lipid substances, totaling 89 species (Figure 6F). The overall level of metabolites in cluster 5 decreased relative to the control, and the analysis results showed that flavonoids were the most abundant, with a total of 64 species (Figure 6G). The content of DAMs belonging to the sixth cluster continued to increase under drought stress, with the highest numbers found for amino acids and their derivatives and alkaloids, with 38 and 33 species, respectively, followed by 17 flavonoids (Figure 6H). These results indicate that the levels of lipid substances may be disturbed during severe drought stress, while amino acids, their derivatives, and alkaloids may be synthesized in large quantities. However, the synthesis of most flavonoids is disrupted, and a small number of flavonoids can still maintain their biosynthesis process free from drought stress.

### 3.6. Enrichment Analysis of Metabolites Under Drought Stress

KEGG enrichment analysis was carried out at various intervals to examine the differential accumulation of DAMs during drought stress. The findings demonstrated that during two days of drought stress, upregulated DAMs were primarily enriched in pathways like cyanoamino acid metabolism, arginine biosynthesis, glycerophospholipid metabolism, and aminoacyl-tRNA biosynthesis. However, downregulated DAMs were significantly enriched in flavonoid and anthocyanin biosynthesis (Figure 7A). Under drought stress for 4 d, downregulated DAMs were significantly enriched in flavonoid biosynthesis, whereas upregulated DAMs exhibited significant enrichment in aminoacyl-tRNA biosynthesis, glucosinolate biosynthesis, biosynthesis of amino acids, and arginine biosynthesis (Figure 7B). Similarly, downregulated DAMs were markedly increased in the production of flavonoids six days after drought stress, while upregulated DAMs showed significant enrichment in aminoacyl-tRNA biosynthesis, biosynthesis of amino acids, glucosinolate biosynthesis, D-amino acid metabolism, 2-oxocarboxylic acid metabolism, cyanoamino acid metabolism, arginine biosynthesis, and glycine, serine, and threonine metabolism (Figure 7C). After existing under drought stress for 8 d, downregulated DAMs were significantly enriched in purine metabolism and nucleotide metabolism. By contrast, upregulated DAMs were primarily enriched in aminoacyl-tRNA biosynthesis, glucosinolate biosynthesis, cyanoamino acid metabolism, biosynthesis of amino acids, and D-amino acid metabolism (Figure 7D). Upregulated DAMs were mainly enriched in linoleic acid metabolism, aminoacyl-tRNA biosynthesis, cyanoamino acid metabolism, glucosinolate biosynthesis, and alpha-linolenic acid metabolism, and the enrichment analysis of downregulated DAMs did not reach a significant level during 10 d of drought stress (Figure 7E). These results collectively indicate that poplars cope with drought stress through temporal metabolic reprogramming, gradually shifting their strategy from the early activation of amino acid and phospholipid metabolism to the dominance of lipid metabolism, while selectively inhibiting high-energy-consuming secondary metabolism.

### 3.7. Combined Analysis of Transcriptome and Metabolome Under Drought Stress in Poplar Trees

Investigations were carried out into the transcriptome and metabolome in tandem to explore the connection between gene expression and metabolites during drought stress. According to our findings, after two days of drought stress, DAMs and DEGs were considerably more abundant in the production of flavonoids and secondary metabolites (Figure 8A). After 4 d of drought stress, DAMs and DEGs showed significant enrichment solely in flavonoid biosynthesis (Figure 8B). After 6 d of drought stress, DAMs and DEGs exhibited significant enrichment in flavonoid biosynthesis, glycine, serine, and threonine metabolism, and biosynthesis of secondary metabolites (Figure 8C). After 8 d of drought stress, DAMs and DEGs were significantly enriched in pathways including glucosinolate biosynthesis, ABC transporters, metabolic pathways, and biosynthesis of secondary metabolites (Figure 8D). After 10 days of drought stress, only the biosynthesis of amino acids and secondary metabolites displayed significant enrichment (Figure 8E). DEGs and DAMs were significantly enriched in the flavonoid biosynthesis pathway during the early-to-middle stages of drought stress (2–6 days), confirming the critical role of flavonoids in plant antioxidant defense. It also suggests that they may be replaced by other metabolic strategies under long-term stress.

### 3.8. Analysis of the Flavonoid Synthesis Pathway Under Drought Stress in Poplars

The previous results showed that the expression levels of genes related to flavonoid synthesis were significantly altered under drought stress, and the content of metabolites was significantly reduced. This data prompted us to conduct further analysis. Firstly, we identified 72 DEGs in the flavonoid synthesis pathway after 2, 4, 6, 8, and 10 d of drought stress, and further calculated the correlation between these genes and flavonoid substances. The findings indicated that 54 DEGs were connected to 52 substances (Figure 9A). Furthermore, the expression levels of these genes and metabolites underwent systematic analysis, and the results showed that 22 out of 54 genes were upregulated, with 4 genes upregulated in the early stage and downregulated in the later stage, and a total of 28 genes were downregulated in expression (Figure 9B). Among the 52 metabolites, 13 were upregulated, 5 exhibited upregulation in the early stage and downregulation in the later stage, and 34 showed downregulation (Figure 9C).

The metabolic pathway diagram illustrates the relationship between gene expression and metabolites. In total, 72 DEGs and 23 DAMs can be annotated onto the flavonoid synthesis pathway (ko00941). The overall contents of metabolites were decreased in the flavonoid synthesis pathway, and levels of gene expression were inhibited. However, there are still members of certain gene families whose expression levels increase with the persistence of drought stress, such as HCT, ANS, and ANR (Figure 9D).

TFs play important regulatory roles in gene expression regulation and substance synthesis. Therefore, the upstream regulatory factors of the flavonoid synthesis pathway were analyzed based on transcriptome and metabolome data. A GRN consisting of 45 TFs and 44 functional genes was constructed, with a total of 131 gene regulatory relationships. Meanwhile, the relationships between 44 functional genes and 41 metabolites were screened (Figure 9E). The top layer contained 20 TFs, of which 19 were upregulated and 1 was downregulated. In the second layer of TFs, there were 25 TFs, of which 23 were upregulated and 2 were downregulated. Among the 44 functional genes identified, 8 genes were upregulated, and only 7 metabolites were upregulated. Among the 45 TFs in the GRN, 6 belonged to the mTERF family, while the C2H2 family and ERF contained 4 genes each. The bHLH, SET, MYB-related, and C3H families each have three members in the GRN.

### 3.9. Analysis of Lipid Metabolic Pathways Under Drought Stress in Poplar Trees

Metabolome analysis results showed that after 10 d of drought stress, lipid substances experienced the highest specific increase compared to the control group (Figure 6F). Therefore, further analysis was conducted on genes involved in lipid metabolism pathways. Based on KEGG information related to lipid metabolism (ko09103), 105 DEGs were screened, of which 83 were upregulated after 10 d of drought stress (Figure 10A). Furthermore, 151 metabolites were screened, of which 96 were upregulated under 10 d of drought stress (Figure 10B). These data suggest that lipids may play a unique role in the response to long-term drought stress in poplar trees.

Similarly, we constructed a GRN consisting of 45 TFs and 40 structural genes, with a total of 154 regulatory relationships. The top layer contained 20 TFs, of which 13 were upregulated (Figure 10C). The second layer contained 25 TFs, of which 21 were upregulated, alongside 40 structural genes. ERF39, GAI, ERF34, DOF5.2, DREB2A, and C3H6 were also present in the flavonoid synthesis pathway GRN, with DREB2A and C3H6 being upregulated genes. This suggests that these TFs may regulate multiple pathways that are involved in drought stress response.

## 4. Discussion

This study systematically revealed the dynamic adaptation mechanisms of *P. davidianain* in response to progressive drought through an integrative analysis of physiological, transcriptomic, and metabolomic data. Key findings indicate the following: (1) Flavonoid metabolism was activated during the early stages of drought but significantly suppressed under severe stress, and lipid metabolism was specifically upregulated during the later stages of stress. (2) The “when one declines, the other rises” pattern of metabolite changes reflects a strategic shift in the plant’s response from “chemical defense” to “physical maintenance”. (3) Gene regulatory network analysis identified six shared TFs (ERF39, GAI, ERF34, DOF5.2, DREB2A, and C3H6) that may serve as key hubs for the coordinated regulation of flavonoid and lipid metabolism. These findings enhance our understanding of the metabolic plasticity underlying drought resistance in woody plants.

Drought is one of the most harmful adverse environmental conditions worldwide. Water deficiency in plants can cause stomatal closure, damage to chloroplasts and photosynthetic organs, and a significant decrease in photosynthesis, while the acceptor side of PSII triggers a cascade of reactions that lead to the formation of O^2−^, which is then catalyzed to form H_2_O_2_ [42]. ROS can serve as a molecular signal to transmit stress signals. However, ROS production and accumulation continue to increase as stress persists. When the number of ROS exceeds a certain threshold, they cause oxidative stress and cell damage [43]. At the same time, the plant’s antioxidant defense response is activated, including both enzymatic and non-enzymatic defense systems, enhancing the plant’s ROS clearance ability. In this study, we observed that, during drought stress, the poplar leaves began to wilt, the chlorophyll content decreased, H_2_O_2_ continued to increase, and antioxidant enzyme activity increased and then decreased. These results demonstrate the process and functioning of the enzymatic defense system, which continued until a decline in enzymatic activity was detected. Transcriptome analysis demonstrated that after 2 d of drought stress, flavonoid biosynthesis, glutathione metabolism, and fatty acid metabolism pathways were significantly enriched. Pathways, including photosynthesis, glycolysis, carbon metabolism, and plant hormone signal transduction, became significantly enriched as drought stress persisted. These results also suggest that during the early phases of drought stress, plants primarily respond to stress through antioxidant pathways, and as the degree of stress increases, photosynthesis and energy metabolism processes are affected. At the same time, plants activate their own defense systems through various hormone pathways.

Plants respond to drought stress by mobilizing their own defense mechanisms to synthesize secondary metabolites [44]. These metabolites include terpenes [45], glutathione [46], ascorbic acid [47], and flavonoids [48], which play a positive role in the plant’s response to drought stress. In this study, DEGs and DAMs exhibited significant enrichment in the flavonoid biosynthesis pathway. Multiple studies have demonstrated that flavonoids play an important role under drought stress conditions. Flavonoids with free radical scavenging activity are capable of alleviating oxidative stress and drought stress in *Arabidopsis* [49]. However, in this study, the accumulation of various flavonoids significantly decreased, which may be due to severe drought stress disrupting the flavonoid synthesis process, as well as the extensive consumption of flavonoids in antioxidant defense processes [50]. It may also be related to the extreme stress intensity in this experiment, possibly as a result of limited photosynthesis, carbon sources, and energy consumption due to severe drought. This phenomenon is considered to be a physiological characteristic of plants, from “active defense” to “maintenance of survival”, which occurs as a result of the failure of the antioxidant system after exceeding the threshold [20]. Alternatively, flavonoid biosynthesis and fatty acid biosynthesis share metabolic precursors. The precursor redistribution mechanism redirects carbon flux from the former to the latter, maintaining membrane structure and energy storage [51]. This reflects the plasticity of poplar metabolism and is a higher-order strategy for its drought resistance.

Lipid molecules are mainly divided into eight categories: glycerides, fatty acyls, sterols, glycerophospholipids, sphingolipids, acrylates, glycolipids, and polyketones [52]. Some lipids, including glycerol phospholipids, such as cardiolipin, phosphatidylethanolamine (PE), and phosphatidylglycerol, serve as key components of biological systems [53]. Under drought stress, the contents of phospholipids and glycolipids in tall fescue increase. This increase enhances membrane fluidity, reduces membrane damage, and thereby improves the plant’s resistance to stress [54]. Some lipids (including phosphatidic acid, phosphatidic acid, and phosphoinositide) also participate in the signal transduction process as signaling molecules [55]. In this study, as drought stress progressed, the majority of genes linked to lipid metabolic pathways also increased. Multiple lipid substances, including LPE, glycerol ester, free fatty acids, and LPC, were significantly induced after 10 d of drought stress. This study reveals that poplars undergo metabolic reprogramming under long-term drought stress, in which upregulation of lipid synthesis pathways is crucial for maintaining cell membrane integrity and ensuring survival.

TFs are crucial regulators of secondary metabolic pathways [56]. This study focused on genes linked to lipid metabolic pathways and flavonoid production, and two multi-level regulatory networks were constructed based on transcriptome data to screen 45 potential upstream regulatory factors. A GRN is a powerful tool for screening core genes; based on the predicted regulatory relationships, key regulatory factors can be excluded to cultivate plants with excellent resistance [57,58]. The functions of homologous genes of some TFs were characterized across other species. For instance, *ERF2* affects the flavonoid content of harvested apples through Ca^2+^ signaling [59]. In the GRN of flavonoid synthesis genes, the largest number comprises mitochondrial transcription termination factors (mTERFs). Following cold stress, the overexpression of *PpmTERF18* in peach fruit can lower the amount of malondialdehyde (MDA), increase proline and phenylalanine ammonia lyase (PAL) activity, enhance the antioxidant capacity, and maintain fruit quality [60]. Arabidopsis mterf6 functional loss mutants exhibit sensitivity to NaCl, mannitol, and abscisic acid (ABA) throughout the development of seedlings and seed germination [61]. However, further research is needed on how mTERF regulates stress response mechanisms, especially the synthesis of flavonoids, in plants. Similarly, 45 potential TFs related to lipid metabolism were screened through the GRN. There have also been studies on the functioning of several genes. For example, in soybeans, *GmMYB73* can downregulate *GL2* and subsequently relieve the inhibition of *PLDα1* expression via *GL2*, promoting lipid accumulation. In transgenic Arabidopsis plants, the expression of *GmMYB73* also increases seed size and thousand-grain weight [62].

Six TFs (ERF39, GAI, ERF34, DOF5.2, DREB2A, C3H6) were screened in the regulatory network of genes related to lipid metabolism and flavonoid synthesis, suggesting that they may exert their effects through multiple biological pathways. Heterologous expression of *EcDREB2A* in tobacco enhances the plant’s reactive oxygen species (ROS) scavenging ability and improves its heat tolerance [63]. *AmDREB2C*, derived from Ammopiptanthus mongolicus, enhances the frost resistance, heat resistance, and drought resistance of Arabidopsis by regulating fatty acid desaturation [64]. DREBs have also been found to regulate the biosynthesis of plant flavonoids [65]. The *AmDREB3* from *A. mongolicus* was transformed into Arabidopsis, which improved its tolerance to drought and high salinities and temperatures. The plant exhibited a purple color and a significant increase in anthocyanins as well as resistance to oxidative stress [66]. These findings imply that some TFs could control plants’ lipid and flavonoid compositions in response to drought stress. The functions and regulatory mechanisms of these TFs deserve further analysis.

## 5. Conclusions

Poplars have important economic and ecological value, yet drought stress severely limits their growth and survival. This study comprehensively integrated physiological, transcriptomic, and metabolomic analyses to systematically reveal the dynamic adaptive mechanisms of *P. davidianain* in response to progressive drought stress. The results demonstrate that drought stress triggers extensive transcriptional reprogramming and metabolic reorganization in poplars. The most significant finding is the drought-induced metabolic shift from “chemical defense” to “physical maintenance”. Flavonoid biosynthesis is significantly inhibited under severe drought, whereas specific lipid metabolites specifically accumulate during later stress stages. Furthermore, multi-level gene regulatory network analysis identified six shared TFs (ERF39, GAI, ERF34, DOF5.2, DREB2A, and C3H6) that may serve as key hubs coordinating flavonoid and lipid metabolic pathways. These findings provide new insights into the metabolic plasticity underlying drought resistance in woody plants and establish a theoretical foundation for the molecular breeding of drought-resistant poplar varieties. The identified key TFs and metabolic biomarkers offer valuable genetic resources for future enhancement of poplar drought tolerance through genetic engineering approaches.

## Figures and Tables

**Figure 1 biology-14-01574-f001:**
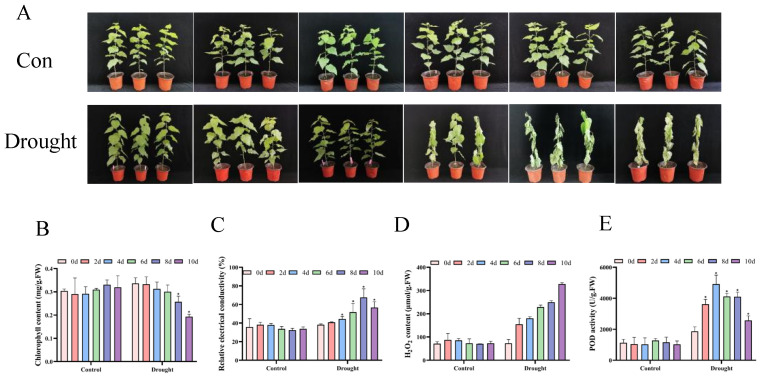
Growth and physiological analysis of *P. davidiana* under drought stress. (**A**) The growth of poplar trees under drought stress at different times; (**B**–**E**) physiological changes in poplar trees under drought stress at different times, including chlorophyll content (**B**), relative conductivity (**C**), H_2_O_2_ content (**D**), and POD activity (**E**). The data are the averages of three independent experiments. Error bars indicate the SD. * The significant difference (*t*-test, *p* <  0.05) compared with 0 d.

**Figure 2 biology-14-01574-f002:**
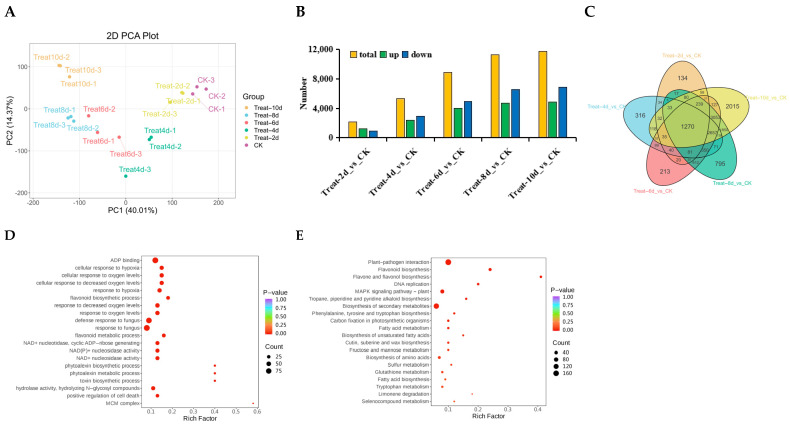
Analysis of *P. davidiana* transcriptomes under various drought conditions. (**A**) PCA analysis between different samples. The degree of dispersion represents the differences between samples. (**B**) Number of DEGs in various samples. (**C**) Venn plot analysis of DEGs under drought stress at different times. (**D**,**E**) GO (**D**) and KEGG (**E**) enrichment analysis of 1270 DEGs shared among all samples.

**Figure 3 biology-14-01574-f003:**
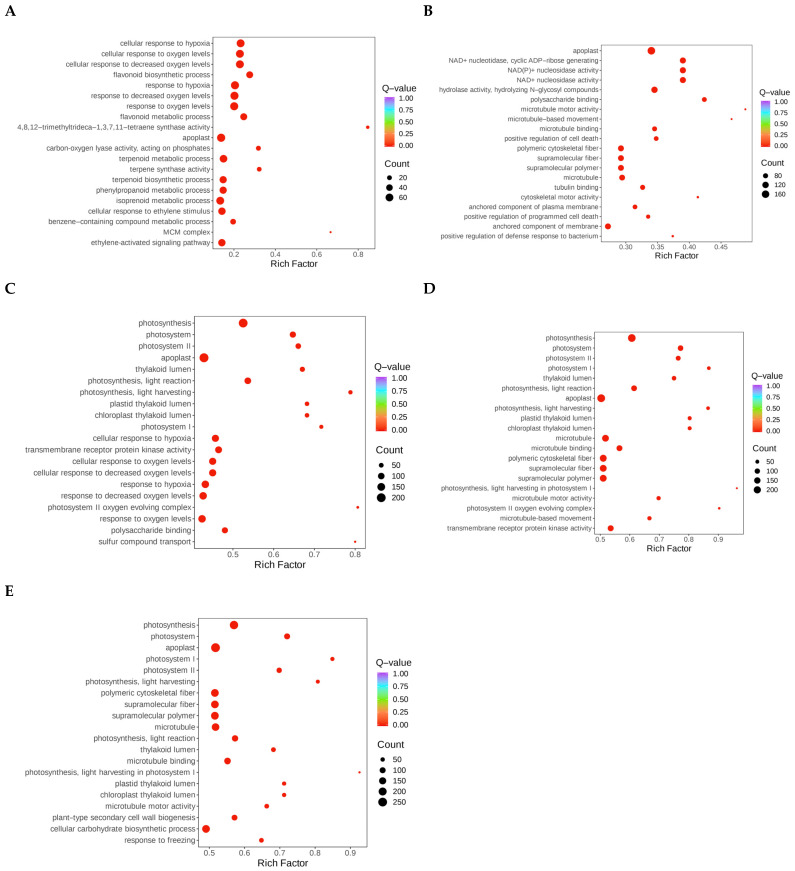
Analysis of DEG GO enrichment during several days of drought stress. (**A**–**E**) GO enrichment analysis of differentially expressed genes after 2 (**A**), 4 (**B**), 6 (**C**), 8 (**D**), and 10 d (**E**) of drought stress. The *p*-value is shown by the color, while the number of DEGs is shown by the area.

**Figure 4 biology-14-01574-f004:**
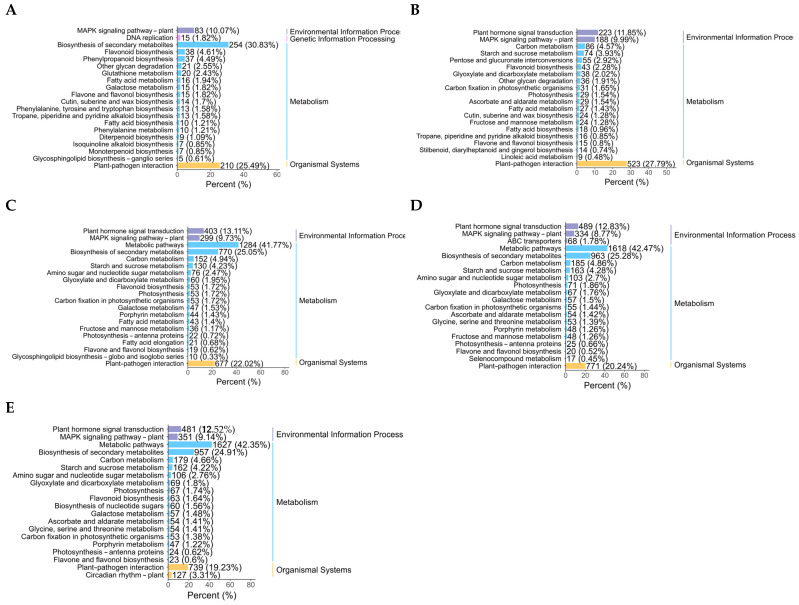
KEGG enrichment analysis of DEGs after exposure to varying levels of drought stress. (**A**–**E**) KEGG enrichment analysis of differentially expressed genes after 2 (**A**), 4 (**B**), 6 (**C**), 8 (**D**), and 10 d (**E**) of drought stress. The colors represent different categories, and the length of the bar chart represents the proportion of gene enrichment.

**Figure 5 biology-14-01574-f005:**
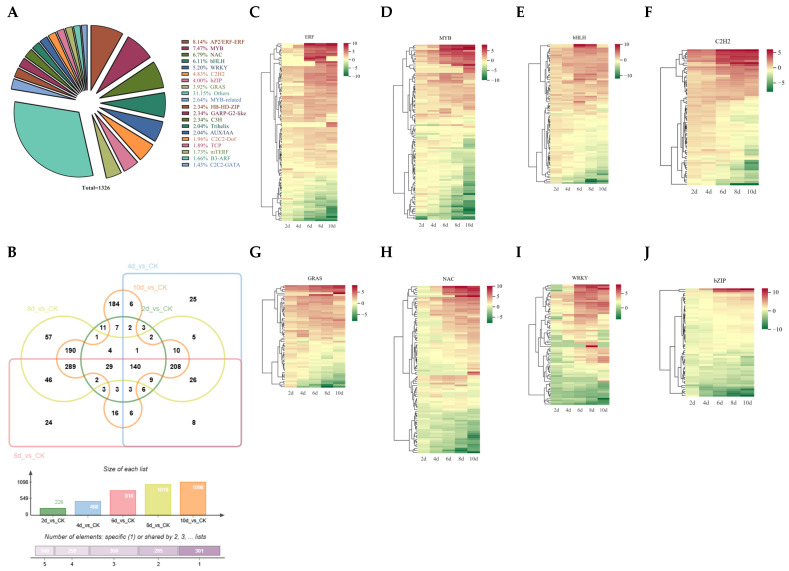
Differential expression analysis of TFs. (**A**) Classification and statistical analysis of differentially expressed TF families. (**B**) Differential expression of TFs in Venn diagram analysis. (**C**–**J**) Analysis of expression patterns of ERF (**C**), MYB (**D**), bHLH (**E**), C2H2 (**F**), GRAS (**G**), NAC (**H**), WRKY (**I**), and bZIP (**J**) family members under drought stress.

**Figure 6 biology-14-01574-f006:**
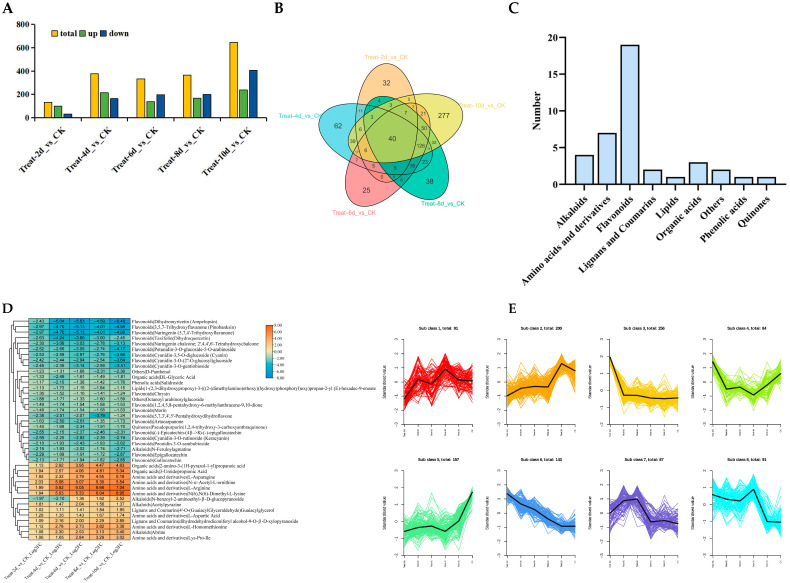
Metabolome analysis of *P. davidiana* under drought stress. (**A**) Statistics of DAMs. (**B**) Venn plot of DAMs under different durations of drought stress. (**C**) Classification and statistics of DAMs shared at all times. (**D**) Analysis of 40 shared DAMs and their expression levels under all drought stress conditions. (**E**) K-means clustering analysis of all DAMs. (**F**,**G**) Classification statistics of DAMs in clusters 3 (**F**), 5 (**G**), and 6 (**H**) for cluster analysis.

**Figure 7 biology-14-01574-f007:**
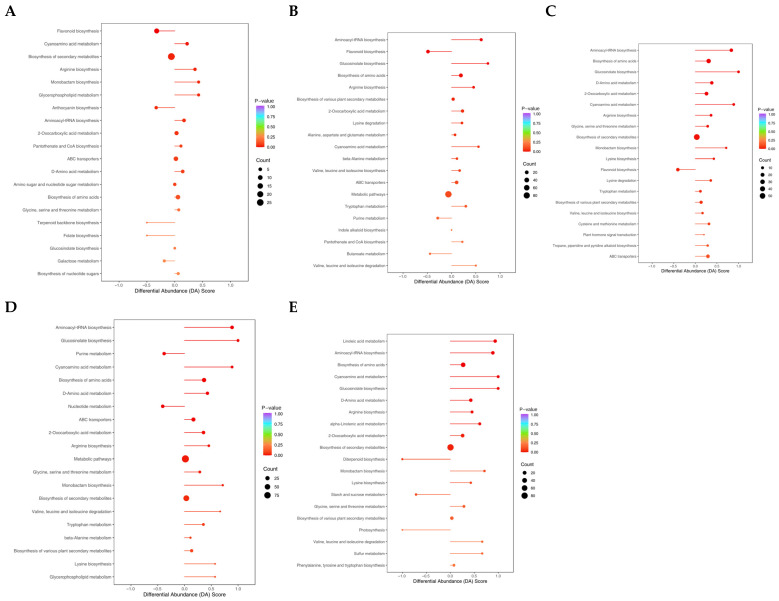
KEGG enrichment analysis of DAMs after various durations of drought stress. (**A**–**E**) KEGG enrichment analysis of DAMs after 2 (**A**), 4 (**B**), 6 (**C**), 8 (**D**), and 10 d (**E**) of drought stress. The direction of the horizontal line represents up- and downregulated DAMs: the dot at the left end of the line represents downregulated, and the right end of the horizontal line represents upregulated DAMs. The vertical axis represents the names of the differential pathways (sorted by *p*-value), and the horizontal axis represents the differential abundance score (DA Score). The DA Score reflects the overall change in metabolites in the metabolic pathway (positive values indicate upregulation and negative values indicate downregulation). The length of the line segment represents the absolute value of the DA Score, the size of the circular dots at the endpoints of the line segment indicates the number of differentially expressed metabolites in the pathway, and the colors of the line segments and dots represent the size of the *p*-value.

**Figure 8 biology-14-01574-f008:**
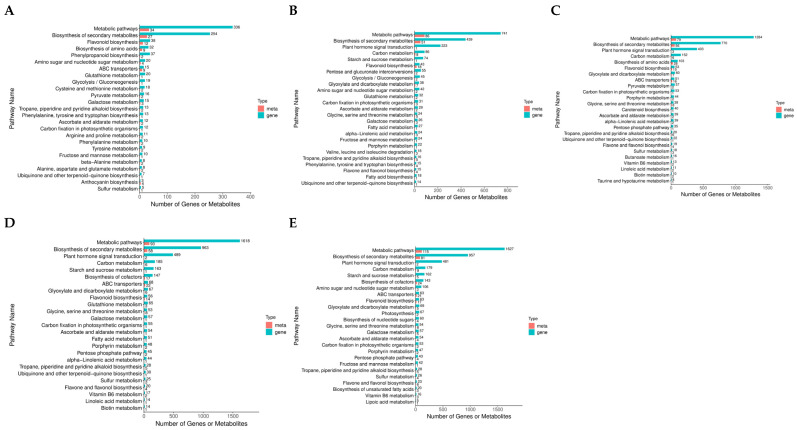
Combined analysis of transcriptome and metabolome. (**A**–**E**) KEGG enrichment analysis of DEGs and DAMs after 2 (**A**), 4 (**B**), 6 (**C**), 8 (**D**), and 10 d (**E**) of drought stress, respectively.

**Figure 9 biology-14-01574-f009:**
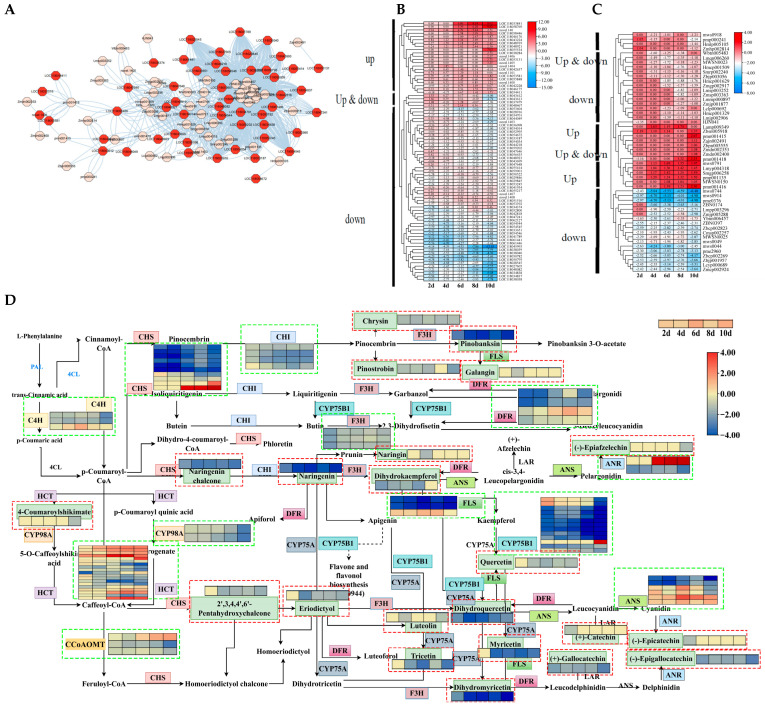
Expression and upstream regulatory factor analysis of DEGs and DAMs in flavonoid synthesis pathways. (**A**) Network diagram of DEGs and DAMs related to flavonoid synthesis. (**B**,**C**) The expression levels of DEGs and DAMs related to flavonoid synthesis. (**D**) Analysis of DEG and DAM expression patterns in flavonoid synthesis pathways. The green box represents DEGs, and the red box represents DAMs. (**E**) GRN of the flavonoid synthesis pathway. The upstream regulatory factors are calculated based on the expression levels of DEGs, while the association between DEGs and DAMs is based on the correlation between expression levels. Different colors represent different layers of distribution.

**Figure 10 biology-14-01574-f010:**
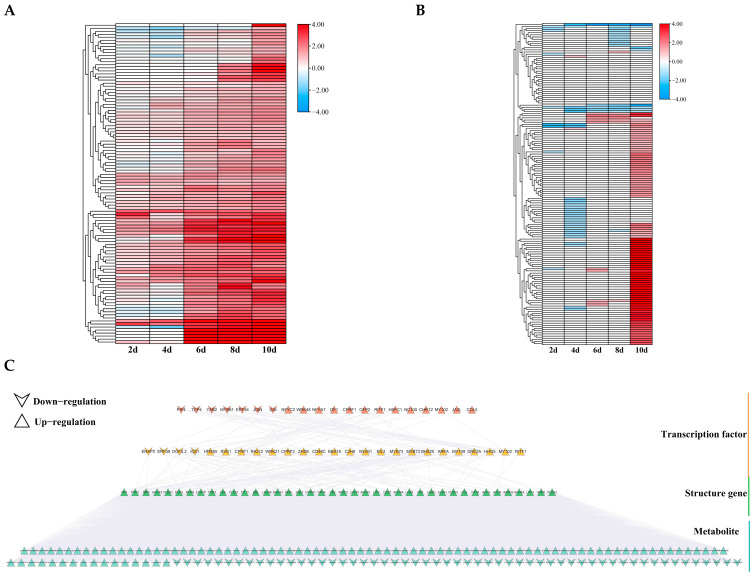
Expression analysis of DEGs and DAMs in the lipid metabolism pathway. (**A**,**B**) Expression levels of DEGs (**A**) and DAMs (**B**) under different drought stress durations. (**C**) GRN of lipid metabolism pathway. Different colors represent different layers of distribution.

## Data Availability

The original contributions presented in this study are included in the article.

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
