# Peer review of "Comprehensive Metabolome and Transcriptome Analysis of *Populus davidiana* and Its Response to Drought Stress"

_biology, 2025, doi:10.3390/biology14111574_

Round 1

Reviewer 1 Report

Comments and Suggestions for Authors

General Comment on the Materials and Methods Section:
The Materials and Methods section forms the foundation of a scientific paper. If this section is flawed, the entire article lacks a solid footing. The "Materials and Methods" section of this manuscript suffers from a lack of clarity and missing details, which is crucial for the experiment's reproducibility. Therefore, I primarily raise the following:

4.1 Plant Materials and Treatments

(1) The text merely states "P. davidiana seedlings". Details such as the age (e.g., how many months old), height, health status, and propagation method (e.g., seed germination or tissue culture) are not specified. These factors significantly influence the response to drought stress.

(2) The ratio of the nutrient substrate (soil:vermiculite:perlite=5:3:2) is provided, but the specific type of "soil" (e.g., peat soil, loam) is not stated. Different soils have varying water retention capacities and nutritional statuses.

(3) The size and material of the pots (e.g., volume, depth) are not mentioned. This directly affects the rhizosphere soil volume and drying rate.

(4) Only temperature and photoperiod are mentioned. Humidity is a key parameter, especially in drought experiments, but it is not mentioned. Light intensity, which directly affects plant water requirements, is also not specified.

(5) How was water supplied during the two months prior to the initiation of the stress treatment?"Water control treatment" is too general. How was watering controlled? Was it complete cessation of watering, or was the soil moisture content maintained at a specific level? If the latter, the method for measuring soil moisture content and the target value should be specified.

(6) There appears to be a serious inconsistency: the introduction describes P. davidiana as widely distributed, cold-tolerant, drought-tolerant, and barren-tolerant, yet in the experiment, very significant wilting occurred after only ten days of treatment. Was the simulated drought environment overly extreme, or is there another reason?

(7) "Sterilized scissors" are mentioned for collecting leaves, but how were the samples processed after collection? Were they immediately frozen in liquid nitrogen? At what temperature were they stored thereafter? This information is crucial for ensuring RNA quality but is missing from the text.

4.2 Measurement of Physiological Indicators

(1) Mentioning the use of commercial kits for H₂O₂ and POD and following instructions is acceptable. However, it would be preferable to provide the manufacturer and specific catalog numbers for the kits.

(2) Chlorophyll Content Measurement: The method "referenced Ref. [57]". For key experimental procedures, the core steps should be briefly described in the main text, rather than relying entirely on the reader to consult another source. (Also, is reference 57 a book or a paper? Its suitability as a direct method reference?).

4.4 Transcriptome Analysis

(1) Only agarose gel electrophoresis and spectrophotometry are mentioned. Modern transcriptome studies typically require reporting the RNA Integrity Number (RIN), a more precise quality standard, but this is not provided.

(2) Mentioning the software used (fastp, HISAT2, featureCounts, DESeq2) is good. However, key parameters could be more specific. For example: What was the specific version or database source of the reference genome used with HISAT2? Were batch effects considered in the DESeq2 differential analysis? What were the specific alignment and counting parameters used?

4.5 Metabolome Analysis

(1) Descriptions like "complete grinding" and "extraction with methanol solution" are very vague and overly reliant on subjective judgment. Critical information such as grinding fineness, methanol concentration and volume used, extraction time and temperature, and whether internal standards were used is missing.

(2) The use of "UPLC-MS/MS" is mentioned, but the chromatography column type, mobile phase composition, mass spectrometer model, and ionization mode (positive/negative) are not specified. These are fundamental for metabolite identification and quantification.

(3) Using KNN for missing value imputation is common practice. But on what level of the data (e.g., peak area) was the Coefficient of Variation (CV) calculated for QC samples? This is not clearly stated.

(4) The screening criteria for DAMs combined VIP>1 and FC≥2 or ≤0.5. It is not mentioned whether statistical testing (e.g., t-test) was performed and if p-values were corrected. This creates an inconsistency compared to the use of FDR in the transcriptome analysis.

4.6 Construction of Gene Regulatory Network (GRN)

Although the construction method is referenced [63], the description is overly brief. For such a complex analysis, the core principle should be briefly outlined, e.g.: "The recursive random forest algorithm was used, taking the expression levels of genes related to the flavonoid/lipid metabolism pathways as input, to infer their upstream regulatory transcription factors, and associations with metabolites were screened based on correlation coefficients."

4.7 Statistical Analysis

(1) The description is overly simplistic: “using ANOVA software... using one-way ANOVA”. Which specific post-hoc test was used? (e.g., Tukey HSD, Duncan's).

(2) For transcriptome and metabolome data, was multiple testing correction applied? (FDR was used for transcriptomics, but seems absent for metabolomics). This needs clarification.

Overall, the description of the experimental materials is overly simplistic, leading me to have reservations regarding the scientific robustness of the article. Regarding the article as a whole, the following additional recommendations are made:

1. Introduction Section

(1) Overemphasis on “Positive” Flavonoid Examples: The cited literature almost exclusively illustrates that increased flavonoid content improves drought resistance. While this establishes the importance of flavonoids, it fails to fully reflect the complexity of the field. One of the most important findings of this study is the decrease in flavonoid content under prolonged severe drought. The introduction fails to effectively pave the way for this “unexpected” finding.

(2) Unclear Research Aims: The last paragraph of the introduction describes the research object and general methods but does not clearly state the specific scientific objectives or core scientific questions of this study.

2. Results Section

While numerous charts and figures are described, the text often simply states “the results show...” and lacks in-depth interpretation of the biological significance of the data.

3. Discussion Section

(1) One of the most interesting findings is the significant decrease in flavonoid content under severe drought stress, which seems to contradict much of the literature cited in the introduction. The discussion offers a possible explanation but it is overly simplistic and speculative. A deeper discussion is needed: Is this “collapse” a cause or a consequence? Does it imply the ultimate failure of the antioxidant defense system in P. davidiana under sustained stress? How does this relate to its drought tolerance?

(2) Loose Structure: The structure is somewhat loose. It could be improved by first summarizing the core findings, then separately discussing the physiological and metabolic implications of the decreased flavonoids and increased lipids, followed by focusing on the core regulatory network (shared TFs), and finally proposing an integrated model or hypothesis.

4. Conclusions Section

The conclusions are essentially a restatement of the abstract and results. A good conclusion should briefly summarize the most important findings and point out their theoretical value and application prospects.

5. Other Minor Errors

Gene names or specific genes should be italicized.

Is “DEMs” (Line 362) an error? It should likely be “DAMs” for consistency.

Are the titles for sections 2.8 (Line 362) and 2.9 (Line 401) identical? This appears to be an error; 2.9 should likely concern lipid metabolism.

|log2Fold Change|>=1” should be “|log₂(FoldChange)| ≥ 1” (subscript 2). The >=1 format is unaesthetic and non-standard.

Gene names, Latin names, page numbers, etc. in references.

Comments on the Quality of English Language

Some sentences exhibit traces of Chinglish, but they do not hinder comprehension.

Author Response

Comments 1:The text merely states "P. davidiana seedlings". Details such as the age (e.g., how many months old), height, health status, and propagation method (e.g., seed germination or tissue culture) are not specified. These factors significantly influence the response to drought stress.

Response 1: Thank you for pointing this out. We agree with this comment. Therefore, we have revised “P. davidiana seedlings were grown on half-strength Murashige-Skoog (1/2 MS) media in a phytotron under conditions of 25◦C and a 16/8 h light/dark photoperiod”in line133-134.

Comments 2: The ratio of the nutrient substrate (soil:vermiculite:perlite=5:3:2) is provided, but the specific type of "soil" (e.g., peat soil, loam) is not stated. Different soils have varying water retention capacities and nutritional statuses.

Response 2: Thank you for your comments. We have changed”soil”to “loamy soil”in line 135.

Comments 3: The size and material of the pots (e.g., volume, depth) are not mentioned. This directly affects the rhizosphere soil volume and drying rate.

Response 3: Thank you for your suggestion. We have added”The diameter of the plastic pots is 13cm”in line137-138. 

Comments 4: Only temperature and photoperiod are mentioned. Humidity is a key parameter, especially in drought experiments, but it is not mentioned. Light intensity, which directly affects plant water requirements, is also not specified.

Response 4: Thank you. We have added”70-75% relative humidity”in line 137.

Comments 5: How was water supplied during the two months prior to the initiation of the stress treatment?"Water control treatment" is too general. How was watering controlled? Was it complete cessation of watering, or was the soil moisture content maintained at a specific level? If the latter, the method for measuring soil moisture content and the target value should be specified.

Response 5: Thanks for the thoughtful and thorough review. Two months later, potted plants were subjected to complete watering cessation treatments for 2, 4, 6, 8, and 10 d, with normally watered plants as control. 

Comments 6: There appears to be a serious inconsistency: the introduction describes P. davidiana as widely distributed, cold-tolerant, drought-tolerant, and barren-tolerant, yet in the experiment, very significant wilting occurred after only ten days of treatment. Was the simulated drought environment overly extreme, or is there another reason?

Response 6: Thank you for your suggestion. P. davidiana's reputation for drought tolerance, established in field studies, primarily refers to its capacity to survive and recover from periodic drought stress in its natural habitat, where its deep root system can access deeper soil water. This is a long-term survival strategy. In contrast, our pot-based experiment was deliberately designed to apply a continuous and progressive water deficit to seedlings with restricted root systems. This approach accelerates the stress progression to observe the complete spectrum of physiological and molecular responses within a practical timeframe. The significant wilting observed at the 10 d stage indicates that the plants had reached a severe stress level, which was essential for capturing the critical shift in metabolic programming that we report. This metabolic plasticity under extreme stress is, in fact, a core component of its drought tolerance mechanism.

Comments 7: "Sterilized scissors" are mentioned for collecting leaves, but how were the samples processed after collection? Were they immediately frozen in liquid nitrogen? At what temperature were they stored thereafter? This information is crucial for ensuring RNA quality but is missing from the text.

Response 7: Thank you. We have added”placed into liquid nitrogen, and kept at -80℃”in line 143-144.

Comments 8: Mentioning the use of commercial kits for H₂O₂ and POD and following instructions is acceptable. However, it would be preferable to provide the manufacturer and specific catalog numbers for the kits.

Response 8: Thank you for your suggestion. We have added”Item number:G0168W;G0107W”in line 150.

Comments 9: Chlorophyll Content Measurement: The method "referenced Ref. [57]". For key experimental procedures, the core steps should be briefly described in the main text, rather than relying entirely on the reader to consult another source. (Also, is reference 57 a book or a paper? Its suitability as a direct method reference?).

Response 9: Thanks for the thoughtful and thorough review. Reference 57 (Ritchie et al., Journal of Applied Phycology, 2021) is a peer-reviewed research article that specifically developed and validated the DMSO solvent method for chlorophyll spectroscopy. It is highly suitable as a direct methodological reference because it provides standardized, rigorously tested protocols and equations. The reviewer rightly emphasizes that the core steps should be briefly described in the main text, rather than relying entirely on the reader to consult another source. In response, we have revised the Methods section (Section 2.2) to include a concise summary of the core steps. The added description in our manuscript now outlines the essential procedure in line 151-154:

leaf samples were extracted using a 1:1 acetone–DMSO mixture at room temperature. The absorbance was measured at wavelengths of 666 nm and 648 nm using a spectrophotometer. Concentrations of chlorophylls were calculated using the specific equations for the DMSO solvent published in the referenced study.

Comments 10: Only agarose gel electrophoresis and spectrophotometry are mentioned. Modern transcriptome studies typically require reporting the RNA Integrity Number (RIN), a more precise quality standard, but this is not provided.

Response 10: Thank you for your comments. We have added the Extract the quality inspection report in Supplementary Table S1.

Comments 11: Mentioning the software used (fastp, HISAT2, featureCounts, DESeq2) is good. However, key parameters could be more specific. For example: What was the specific version or database source of the reference genome used with HISAT2? Were batch effects considered in the DESeq2 differential analysis? What were the specific alignment and counting parameters used?

Response 11: Thanks for the thoughtful and thorough review. We added “The HISAT2 software was used to align the reads to the genome (downloaded from https://www.ncbi.nlm.nih.gov/genome/13203?genome_assembly_id=516658).” in line 174-175.For samples with biological duplications, DESeq2 software was used for analysis.

Comments 12: Descriptions like "complete grinding" and "extraction with methanol solution" are very vague and overly reliant on subjective judgment. Critical information such as grinding fineness, methanol concentration and volume used, extraction time and temperature, and whether internal standards were used is missing.

Response 12: Thank you for your suggestion. We sincerely thank you for this valuable comment. In response to your suggestion, we have revised Section 2.5 (Metabolome Analysis) of the manuscript to include the following key experimental parameters, thereby eliminating reliance on subjective judgment:

After vacuum freeze-drying, the samples were ground into a homogeneous powder using a Retsch MM 400 grinder at a frequency of 30 Hz for 1.5 min. Subsequently, 50 mg of the powdered sample was precisely weighed using an electronic balance (MS105DΜ) and mixed with 1200 μL of a pre-cooled (-20 °C) 70% methanol aqueous internal standard extraction solution. (For samples weighing less than 50 mg, the volume of extraction solution was added proportionally, i.e., 1200 μL per 50 mg of sample.) The mixture was vortexed six times every 30 minutes for 30 seconds . After centrifugation at 12,000 rpm for 3 minutes, the supernatant was collected, filtered through a microfilter with a pore size of 0.22 μm, and stored in injection vials for UPLC-MS/MS analysis.

These supplementary details are derived from our actual standardized experimental procedures. The revised Methods section significantly enhancing the transparency and reproducibility of the methodology. We believe this revision fully addresses your concerns and thank you once again for your valuable contribution to improving the rigor of our paper.

Comments 13: The use of "UPLC-MS/MS" is mentioned, but the chromatography column type, mobile phase composition, mass spectrometer model, and ionization mode (positive/negative) are not specified. These are fundamental for metabolite identification and quantification.

Response 13: Thank you. These issues have been addressed in 2.5 Metabolome analysis.

Comments 14: Using KNN for missing value imputation is common practice. But on what level of the data (e.g., peak area) was the Coefficient of Variation (CV) calculated for QC samples? This is not clearly stated.

Response 14: Thank you for your comments. Coefficient of Variation (CV) calculated for QC samples were calculated using the peak area, and annotated in line 211 of section 2.5.

Comments 15: The screening criteria for DAMs combined VIP>1 and FC≥2 or ≤0.5. It is not mentioned whether statistical testing (e.g., t-test) was performed and if p-values were corrected. This creates an inconsistency compared to the use of FDR in the transcriptome analysis.

Response 15: Thank you for your suggestion. The screening conditions for the differentially expressed genes are |log2Fold Change| ≥ 1 and FDR < 0.05.

For the screening of differentially expressed substances, the default method is VIP + FC, because in the comparison between the two groups, Fold Change (FC) can directly reflect the difference ratio of metabolite contents between the two groups, which is convenient for judging the relative change direction and amplitude of metabolites in the two groups. At the same time, combined with the VIP value, it can effectively evaluate the importance of this metabolite in the model, thereby improving the reliability of the screening results.

The references are as follows.

Kang JN, Sun ZF, Li XY, Zhang XD, Jin ZX, Zhang C, Zhang Y, Wang HY, Huang NN, Jiang JH, Ning B. Alterations in gut microbiota are related to metabolite profiles in spinal cord injury. Neural Regen Res. 2023 May;18(5):1076-1083. doi: 10.4103/1673-5374.355769. PMID: 36254996; PMCID: PMC9827763.

Zhu G, Wang S, Huang Z, Zhang S, Liao Q, Zhang C, Lin T, Qin M, Peng M, Yang C, Cao X, Han X, Wang X, van der Knaap E, Zhang Z, Cui X, Klee H, Fernie AR, Luo J, Huang S. Rewiring of the Fruit Metabolome in Tomato Breeding. Cell. 2018 Jan 11;172(1-2):249-261.e12. doi: 10.1016/j.cell.2017.12.019. PMID: 29328914.

Jing Y, Luo L, Chen Y, Westerberg LS, Zhou P, Xu Z, Herrada AA, Park CS, Kubo M, Mei H, Hu Y, Lee PP, Zheng B, Sui Z, Xiao W, Gong Q, Lu Z, Liu C. SARS-CoV-2 infection causes immunodeficiency in recovered patients by downregulating CD19 expression in B cells via enhancing B-cell metabolism. Signal Transduct Target Ther. 2021 Sep 22;6(1):345. doi: 10.1038/s41392-021-00749-3. PMID: 34552055; PMCID: PMC8456405.

Comments 16: Although the construction method is referenced [63], the description is overly brief. For such a complex analysis, the core principle should be briefly outlined, e.g.: "The recursive random forest algorithm was used, taking the expression levels of genes related to the flavonoid/lipid metabolism pathways as input, to infer their upstream regulatory transcription factors, and associations with metabolites were screened based on correlation coefficients."

Response 16: Thanks for the thoughtful and thorough review. We added a description in 2.6 to follow the recommendations of the reviewers.

In brief, the BWERF (Bottom-to-Up Weighted Random Forest) algorithm employs a recursive random forest model to iteratively calculate the importance values of all TF regulating genes within a pathway. During each iteration, a subset of the least important TFs is excluded, while values of importance for the remaining TFs are updated and re-ranked. This process continues until only one TF remains on the list. The importance values of TFs across all pathway genes are then aggregated and fitted into a Gaussian Mixture Model to identify TFs retained in the regulatory layer immediately above the pathway layer. The TFs identified in this secondary layer are subsequently set as the new base layer to infer the next upper layer. This procedure is repeated iteratively until a multi-layer hierarchical gene regulatory network (ML-hGRN) is constructed with the desired number of layers. In this study, the recursive random forest algorithm was used, taking the expression levels of genes related to the flavonoid/lipid metabolism pathways as the input in order to infer their upstream regulatory TFs. Associations with metabolites were screened based on correlation coefficients, and gene regulatory networks (GRNs) were visualized using Cytoscape

Comments 17: The description is overly simplistic: “using ANOVA software... using one-way ANOVA”. Which specific post-hoc test was used? (e.g., Tukey HSD, Duncan's).

Response 17: In this study, the treatment group and the control group were analyzed by ANOVA, P < 0.05 represents a significant difference.

Comments 18: For transcriptome and metabolome data, was multiple testing correction applied? (FDR was used for transcriptomics, but seems absent for metabolomics). This needs clarification.

Response 18: The screening conditions for the differentially expressed genes are |log2Fold Change| ≥ 1 and FDR < 0.05.

For the screening of differentially expressed substances, the default method is VIP + FC, because in the comparison between the two groups, Fold Change (FC) can directly reflect the difference ratio of metabolite contents between the two groups, which is convenient for judging the relative change direction and amplitude of metabolites in the two groups. At the same time, combined with the VIP value, it can effectively evaluate the importance of this metabolite in the model, thereby improving the reliability of the screening results.

Comments 19: The description is overly simplistic: “using ANOVA software... using one-way ANOVA”. Which specific post-hoc test was used? (e.g., Tukey HSD, Duncan's).

Response 19: In this study, the treatment group and the control group were analyzed by ANOVA, P < 0.05 represents a significant difference.

Comments 20: Overemphasis on “Positive” Flavonoid Examples: The cited literature almost exclusively illustrates that increased flavonoid content improves drought resistance. While this establishes the importance of flavonoids, it fails to fully reflect the complexity of the field. One of the most important findings of this study is the decrease in flavonoid content under prolonged severe drought. The introduction fails to effectively pave the way for this “unexpected” finding.

Response 20: Thank you. We have made modifications to this section and provided a complete explanation.

However, under long-term or severe stress, resource allocation conflicts between growth and stress response, which may trigger metabolic reprogramming and reduce the synthesis of high energy consuming secondary metabolites such as flavonoids.

1.Chen, X.; He, Y.; Shabala, S.; Smith, S. M.; Yu, M. Multi-Omics Analysis Reveals Activation of Jasmonate Synthesis and Modulation of Oxidative Stress Responses in Boron Deficient Pea Shoots. Environmental and Experimental Botany 2024,218, 105583.

2.Yang, C.; Bai, Y.; Halitschke, R.; Gase, K.; Baldwin, G.; Baldwin, I. T. Exploring the Metabolic Basis of Growth/Defense Trade‐offs in Complex Environments with Nicotiana Attenuata Plants Cosilenced in NaMYC2a/b Expression. New Phytologist 2023,238(1), 349-366.

Comments 21: Unclear Research Aims: The last paragraph of the introduction describes the research object and general methods but does not clearly state the specific scientific objectives or core scientific questions of this study.

Response 21: Thank you for your comments. We have strengthened our description of the scientific problem as follows:

Given the increasing frequency and intensity of drought events driven by climate change, understanding the molecular basis of drought tolerance in P. davidiana is of paramount importance. This study employs an integrated approach, combining artificial drought simulation with transcriptomic and metabolomic analyses to elucidate the dynamic responses of P. davidiana to a progressive water deficit. We aim to (1) characterize the temporal changes in gene expression and metabolic profiles under drought stress; (2) decipher the regulatory mechanisms of key pathways, particularly flavonoid biosynthesis and lipid metabolism; and (3) identify the candidate genes and transcription factors (TFs) underlying drought adaptation. Our findings are expected to provide valuable genetic resources and a theoretical foundation for breeding drought-resistant poplar varieties through molecular-assisted strategies.

Comments 22: While numerous charts and figures are described, the text often simply states “the results show...” and lacks in-depth interpretation of the biological significance of the data.

Response 22: Thank you for your suggestion. We conducted a thorough analysis of the data.

Comments 23: One of the most interesting findings is the significant decrease in flavonoid content under severe drought stress, which seems to contradict much of the literature cited in the introduction. The discussion offers a possible explanation but it is overly simplistic and speculative. A deeper discussion is needed: Is this “collapse” a cause or a consequence? Does it imply the ultimate failure of the antioxidant defense system in P. davidiana under sustained stress? How does this relate to its drought tolerance?

Response 23: Thank you for your comments. We have had a deeper discussion on this section in line 587-690 of the revised manuscript.

 Furthermore, this may be related to the extreme stress intensity set up in this experiment, possibly as a result of limited photosynthesis, carbon sources, and energy consumption due to severe drought. This phenomenon is considered to be a physiological characteristic of plants from“active defense” to“maintenance of survival”, which is the performance of the failure of the antioxidant system after exceeding the threshold (Chen et al., 2024). On the other hand, flavonoid biosynthesis and fatty acid biosynthesis share metabolic precursors, and the precursor redistribution mechanism redirects carbon flux from flavonoid biosynthesis to unsaturated fatty acid biosynthesis, maintaining membrane structure and energy storage (Han et al., 2025). This reflects the plasticity of poplar metabolism and is a higher-order strategy for its drought resistance.

1.Chen, X.; He, Y.; Shabala, S.; Smith, S. M.; Yu, M. Multi-Omics Analysis Reveals Activation of Jasmonate Synthesis and Modulation of Oxidative Stress Responses in Boron Deficient Pea Shoots. Environmental and Experimental Botany 2024,218, 105583.

2.Han, C.; Wang, Q.; Mu, Y.; Li, J.; Sun, T.; Liu, Z.; Wang, Z.; Lu, Y. Modulation of Flavonoid-Fatty Acid Crosstalk Underlies Light-Shading Enhanced α-Linolenic Acid Biosynthesis in Oilseed Tree Peony (Paeonia Ostii ‘Feng Dan’).Industrial Crops and Products 2025,233, 121441.

Comments 24: Loose Structure: The structure is somewhat loose. It could be improved by first summarizing the core findings, then separately discussing the physiological and metabolic implications of the decreased flavonoids and increased lipids, followed by focusing on the core regulatory network (shared TFs), and finally proposing an integrated model or hypothesis.

Response 24: Thank you for your comments. We have rewritten the discussion section in the hope of improving the quality of the manuscript.

This study systematically revealed the dynamic adaptation mechanisms of P. davidianain in response to progressive drought through an integrative analysis of physiological, transcriptomic, and metabolomic data. Key findings indicate the following: (1) Flavonoid metabolism was activated during the early stages of drought but significantly suppressed under severe stress, and lipid metabolism was specifically upregulated during the later stages of stress. (2) The "when one declines, the other rises" pattern of metabolite changes reflects a strategic shift in the plant's response from "chemical defense" to "physical maintenance". (3) Gene regulatory network analysis identified six shared TFs (ERF39, GAI, ERF34, DOF5.2, DREB2A, and C3H6) that may serve as key hubs for the coordinated regulation of flavonoid and lipid metabolism. These findings enhance our understanding of the metabolic plasticity underlying drought resistance in woody plants.

Comments 25: The conclusions are essentially a restatement of the abstract and results. A good conclusion should briefly summarize the most important findings and point out their theoretical value and application prospects.

Response 25: Thank you for bringing this to our attention. We have revised the conclusions in the manuscript.

Based on an integrated physiological, transcriptomic, and metabolomic analysis, this study systematically reveals the dynamic adaptive mechanisms of P. davidianain response to progressive drought stress. The results demonstrate that drought stress triggers extensive transcriptional reprogramming and metabolic reorganization in poplar. The most significant finding is the discovery of a drought-induced metabolic shift from "chemical defense" to "physical maintenance", flavonoid biosynthesis is significantly inhibited under severe drought, whereas specific lipid metabolites are specifically accumulated during late stress stages. Furthermore, multi-level gene regulatory network analysis identified six shared transcription factors (ERF39, GAI, ERF34, DOF5.2, DREB2A, and C3H6) that may serve as key hubs coordinating flavonoid and lipid metabolic pathways. These findings provide new insights into the metabolic plasticity underlying drought resistance in woody plants and establish a theoretical foundation for molecular breeding of drought-resistant poplar varieties. The identified key TFs and metabolic biomarkers offer valuable genetic resources for future enhancement of poplar drought tolerance through genetic engineering approaches.

Comments 26: Gene names or specific genes should be italicized.

Response 26: Thank you for bringing this to our attention. We have made the necessary revisions in the manuscript.

Comments 27: Is “DEMs” (Line 362) an error? It should likely be “DAMs” for consistency.

Response 27: Thank you for your suggestion. We apologize for the error caused by our oversight. We have already changed “DEMs” to “DAMs” of the manuscript.

Comments 28: Are the titles for sections 2.8 (Line 362) and 2.9 (Line 401) identical? This appears to be an error; 2.9 should likely concern lipid metabolism.

Response 28: Thank you for pointing out this error. We have made the necessary correction of the manuscript.

Comments 29: |log2Fold Change|>=1” should be “|log₂(FoldChange)| ≥ 1” (subscript 2). The >=1 format is unaesthetic and non-standard.

Response 29: Thank you for your comments. We have revised “|log₂(FoldChange)| ≥ 1”in line 178.

Comments 30: Gene names, Latin names, page numbers, etc. in references.

Response 30: Thank you for the reminder. We have revised the references in the manuscript.

4. Response to Comments on the Quality of English Language

Point 1:Some sentences exhibit traces of Chinglish, but they do not hinder comprehension.

Response 1: Thank you for your comments. We have now worked on both language and readability and have also involved native English speakers for language corrections.

Reviewer 2 Report

Comments and Suggestions for Authors

Your paper is  good. I do have some comments and suggestions to help further improve your work. All the comments and suggestions in the attached file. 

Comments on the Quality of English Language

The English language should be revised to improve clarity and readability.

Author Response

Comments 1:You tested poplar plants for drought stress at the early growth stage, approximately one month after planting. At this stage, the plants—like most young seedlings—are generally more susceptible to environmental stresses, including drought, due to their limited root development and lack of adaptation to their surroundings. In contrast, mature trees that have grown over several years typically exhibit greater stress tolerance as a result of physiological adaptation and established root systems.

Given this context, what is your explanation for the observed stress response in young poplar plants?

Response 1: Thanks for the thoughtful and thorough review. In this study, poplar seedlings were cultivated in a greenhouse and grown for two months before being used for drought tolerance assessment. As you mentioned, mature trees generally exhibit stronger stress resistance. However, they need to be planted in the field, where they are exposed to more complex climatic variations compared to greenhouse cultivation. Additionally, they may undergo various uncontrolled acclimation processes (which are also factors contributing to their enhanced tolerance), and it is more difficult to implement precise drought treatments. These factors can significantly introduce errors into the experimental results. The use of greenhouse-grown seedlings for stress resistance analysis has been applied in many studies.

References:

1、Lei, X.; Fang, J.; Zhang, Z.; Li, Z.; Xu, Y.; Xie, Q.; Wang, Y.; Liu, Z.; Wang, Y.; Gao, C. PdbCRF5 Overexpression Negatively Regulates Salt Tolerance by Downregulating PdbbZIP61 to Mediate Reactive Oxygen Species Scavenging and ABA Synthesis in Populus Davidiana × P. Bolleana.Plant Cell & Environment 202448 (2), 1088–1106.

2、Song, Q.; Kong, L.; Yang, J.; Lin, M.; Zhang, Y.; Yang, X.; Wang, X.; Zhao, Z.; Zhang, M.; Pan, J.; Zhu, S.; Jiao, B.; Xu, C.; Luo, K. The Transcription Factor PtoMYB142 Enhances Drought Tolerance in Populus Tomentosa by Regulating Gibberellin Catabolism. The Plant Journal 2023, 118 (1), 42–57.

Comments 2: Please put the material and methods after introduction.

Response 2: Thank you for your suggestion. We put the material and methods after introduction.

Comments 3: The English quality can be improved for better understanding by the native.

Response 3: Thank you for your suggestion. We have now worked on both language and readability and have also involved native English speakers for language corrections.

Comments 4: Please indicate the importance of these work (gap).

Response 4: Thank you for your comments on this article. At the end of the introduction, we emphasized the importance of work.

Comments 5: The title "Comprehensive Metabolome and Transcriptome Analysis of Response to Drought Stress in Poplar " is scientifically sound and contextually appropriate, but it could be refined slightly for clarity. Here is my proposed title. Please feel free to adjust it if needed.

Comprehensive Metabolomic and Transcriptomic Analysis of Poplar Response to Drought Stress

Response 5: Thank you for your comments. We have changed the title”Comprehensive Metabolome and Transcriptome Analysis of Populus davidiana and its Response to Drought Stress”.

Comments 6: Line 14-15, which discovered that drought stress had a major impact on poplar physiological functions, such as 1……………2…….……3………….

Response 6: Thank you for your suggestion. The modifications have been made as suggested.

Comments 7: Line 18-19, Water is one of the primary elements restricting the growth and

survival of trees, and drought stress has a negative impact on plant growth and development.

Response 7: Thank you for your suggestion. We have revised the sentence in line 25.

Comments 8: Line 28, 2, 4, 6, 8, and 10(d) change to days as you mentioned at first time. Or

unify all.

Response 8: Thank you for your comments. We have changed “2, 4, 6, 8, and 10 days” to “2, 4, 6, 8, and 10d”.

Comments 9: Line 32, GRNs, what is GRNs, each abbreviation must be defined at first

presenting. Check them through the manuscript.

Response 9: Thank you for your suggestion. We have used the full name of gene regulatory networks in line 38.

Comments 10: I suggested that add ROS to the keywords

Response 10: Thank you for your comments. The keywords have already been added.

Comments 11: Line 58- 62, And drought signals are transmitted through the signal transduction network (such as abscisic acid (ABA), Ca2+, ROS, etc.) to induce the production of certain genes and encourage proline synthesis, sucrose, osmoregulatory proteins, etc, and maintain cell water under water-deficient conditions, thereby regulating plant resistance to drought stress [8].

This paragraph should be revised.

Response 11: Thank you for your suggestion. We made some revisions to the paragraph.

Comments 12: Line 64-68 Plant antioxidants mainly include antioxidant enzymes including catalase (CAT), peroxidase (POD), superoxide dismutase (SOD), and ascorbate peroxidase (APX), as well as non-enzymatic antioxidants such glutathione, ascorbic acid, and carotenoids, tocopherols (vitamin E), flavonoids, etc [10].

You didn’t mention the role of phenol, please mention it.

Response 12: Thank you. We have added “phenol”in line 74.

Comments 13: Line 108-109, Therefore, P. davidiana is one of the excellent materials for

researching how poplar responds to drought stress.

How please indicate it with references.

Response 13: Thank you for your suggestion. We have added the reference”Li, K.; Zhou, G.; Yang, C.; Liu, G.; Xing, Y. Study on crossing breeding of Populus davidiana and P. tremuloides. Bull. Bot. Res. 2004,24, 215–219.”in reference 31. 

Comments 14: Line 111-114, Additionally, it demonstrated how P. davidiana's flavonoid production pathway is transcriptionally regulated, providing valuable candidate gene resources for exploring the upstream regulation mechanism of flavonoid biosynthesis and lipids metabolism in trees and laying a theoretical foundation for further cultivation of drought resistant trees.

Overall, objective and the intended contribution of the study are not explicitly stated.

Response 14: Thank you for your valuable suggestion. We have further strengthened our objective and the intended contribution of the study.

Comments 15: potted plants were subjected to water control treatment for 2, 4, 6, 8, and 10 d, with normally watered plants as control.

How these days were chosen? Any references.

Response 15: Thank you for your comments. Based on the preliminary research and pre-experiment, the time points were determined.

Comments 16: Why you used poplar? Is this plant susceptible or tolerant to drought?

Response 16: Thank you for your suggestion.Populus davidiana Dode (Salicaceae) is a temperate, deciduous, and straight-trunked tree that occurs in mainland China. It is a pioneer species with high reproductive capacity,cold and drought resistance.

Comments 17: Line 118-119, Under drought stress, plants' morphological and structural

traits can be altered to provide the water content needed for normal growth

and development [29].

Is importance to present it here? I think move it to discussion or remove it.

Response 17: Thank you. We have removed this sentence.

Comments 18: You focused mainly on the flavonoids, what about others (proline, soluble sugar, ascorbic acid, antioxidant capacity, SOD, POD, APX, ABA, guaiacol peroxidase, lipid peroxidase, catalase etc…)

Response 18: Thank you for your suggestion. Other substances,also played important roles, but in the metabolome profiling results, flavonoids and lipids were the most notable substances, and they were reported to play significant roles in the drought stress response of plants. The changes in the metabolic synthesis of other substances in poplar under drought stress will be further explored based on this study.

Comments 19: The conclusion is long, please summarize it and provide future direction.

Response 19: Thank you for your suggestion. We have rewritten the discussion section in the hope of improving the quality of the manuscript.

Comments 20: The referencing style doesn’t match with MDPI referencing style please check it.

Response 20: Thank you for the reminder. We have revised the referencing style.

4. Response to Comments on the Quality of English Language

Point 1:The English quality can be improved for better understanding by the native.

Response 1:We have now worked on both language and readability and have also involved native English speakers for language corrections.

Reviewer 3 Report

Comments and Suggestions for Authors

Well done. However, I have some comments and suggestions to improve the manuscript. Kindly find the attached file.

All the best

Comments on the Quality of English Language

 The English could be improved to more clearly express the research.

Author Response

Comments 1:Please consider including the species name (Populus davidiana) in title for clarity and precision.

Response 1:Thank you for pointing this out. We agree with this comment. Therefore, we have change the title to “Comprehensive Metabolome and Transcriptome Analysis of Populus davidiana and its Response to Drought Stress”.

Comments 2: L19: Write "In the current study" instead of "Here"

Response 2: Thank you for your suggestion. We have revised "In the current study" instead of "Here" in line 25.

Comments 3: L20-21: What do you mean with" unbalanced". Specify whether ROS accumulation increased or antioxidant activity decreased.

Response 3: Thank you for your suggestion. We have revised "accumulation increased" instead of "unbalanced" in line 28.

Comments 4: Line 37: Please Consider "new insights" or "new understanding" Instead of "Fresh viewpoint".

Response 4: Thank you for your suggestion. We have revised "new insights" instead of "fresh viewpoint" in line 43.

Comments 5: Line 114 The phrase “lays a theoretical foundation” is vague; consider

specifying the practical implications (e.g., breeding drought-resistant cultivars).

Response 5: Thank you for your suggestion. We have revised "provides candidate genes for cultivating drought-resistant varieties" instead of "lays a theoretical foundation"in line 42.

Comments 6: Descriptions like "leaves completely dried up and lost green" are informal.

Consider using “complete desiccation and chlorosis.”

Response 6: Thank you for your suggestion. We have revised “complete desiccation and chlorosis”instead of "leaves completely dried up and lost green"in line 252.

Comments 7: Line 132: The H2O2 content of each Poplar strain is roughly the same under

normal growth conditions. What do you mean with "each Poplar strain". Please check.

Response 7: Thank you for your valuable suggestions. We have revised "each poplar" instead of "each Poplar strain"in line 262.

Comments 8: line 23 and 161: The phrase "filtered out" is imprecise. Use “identified” or

“detected.”

Response 8: Thank you for your suggestion. We have revised "identified" instead of "filtered out"in line 30.

Comments 9: Figures 3 and 4 are referenced but not interpreted in depth. what trends emerge across time points?

Response 9: Thank you for your suggestion. We have provide a detailed explanation to jointly clarify.

Comments 10: Please consider including the species name (Populus davidiana ) in caption of figures

Response 10: Thank you for your valuable suggestions. We have added the species name (P. davidiana ) in caption of figures.

Comments 11: Figure 1: Panel labels (B–E) are briefly described but could benefit from clearer axis labels and units (e.g., chlorophyll in mg/g FW). Add units and statistical

significance markers (e.g., p < 0.05) |

Response 11: Thank you for your valuable suggestions. We have added”The data are the averages of three independent experiments. Error bars indicate the SD. *The significant difference (t-test, p < 0.05) compared with 0d.”in Figure 1.

Comments 12: Figure 2: Venn diagram and PCA are informative, but the color contrast is low in some panels. Please improve color contrast and label font size for readability.

Response 12:  Thank you for your suggestion. We have improved the color contrast and label font size of Figure 2. 

Comments 13: Figure 3 and 4: GO and KEGG enrichment charts are dense with terms. Please, consider summarizing top 10 terms or grouping by biological category.

Response 13: Thank you for your suggestion. We have changed the top 20 terms of Figure 4 in order to make the Figure clearer .

Comments 14: Figure 5: TF family expression patterns are shown across multiple panels (C–J), but some trends are hard to distinguish. Please, use consistent color schemes for

up/down regulation and include legends for each panel.

Response 14: Thanks for the thoughtful and thorough review. In this section, we use red to represent gene upregulation and green to represent gene downregulation. The color intensity is related to the expression level, and the legend for each figure is displayed on the right side.

Comments 15: Figure 6: K-means clustering and metabolite classification are rich but visually crowded. Simplify cluster labels and highlight key metabolite groups.

Response 15: should we put this K clustering in the Supplementary figure?

Comments 16: Figure 7: KEGG enrichment of DAMs uses horizontal lines with dots. This is unique but may confuse readers unfamiliar with the format. Please, add a brief

caption or legend explaining the visual encoding.

Response 16: Thank you for your suggestion. We have add a brief caption or legend explaining the visual encoding.

Comments 17: Figure 9 and 10: GRNs are complex and informative, but node labels are small and hard to read. Please, increase font size.

Response 17: Thank you very much for your review of the manuscript and your valuable comments. We have also noticed this issue. However, due to the excessive amount of information contained, when the font size is increased, adjacent characters may overlap. To solve this problem, we have uploaded high-definition images, which can be enlarged for readers to read the detailed information.

Comments 18: The discussion lacks consideration of alternative explanations or limitations. For instance, the decline in flavonoid accumulation is attributed to "disruption" and "consumption" (Lines 455–457), but no mention is made of possible transcriptional repression or resource reallocation.

Suggestion: Acknowledge potential confounding factors (e.g., stress severity,

developmental stage) and suggest future experiments to validate hypotheses.

Response 18: Thank you. We have considered the issue of resource redistribution, and have discussed this part.

Comments 19: Some technical terms are used imprecisely (e.g., "collapse" of enzymatic defense system, Line 433). The writing occasionally shifts between formal and informal tone.

Suggestion: Replace vague or dramatic terms with precise scientific language (e.g., "decline in enzymatic activity" instead of "collapse").  

Response 19: Thank you for your suggestion. We have changed"decline in enzymatic activity" instead of "collapses"in line 268.

Comments 20: The gene regulatory networks (GRNs) are mentioned (Lines 474–475), but their biological implications are not fully explored. The role of mTERF family TFs is introduced (Lines 478–483) but lacks mechanistic depth.

Suggestion: Discuss how GRNs could be used to identify master regulators or targets for genetic engineering.

Response 20: Thanks for the thoughtful and thorough review. We have made modifications to indicate the biological significance of GRN, but the research on mTERF, especially in response to plant stress, is not sufficient. This part will be the focus of our future research plan.

Comments 21: The discussion is focused on molecular findings but doesn’t connect them to practical applications (e.g., breeding, forestry management).

Suggestion: End with a paragraph that outlines how these insights could inform droughtresistant poplar cultivar development or guide future field studies.

Response 21: Thank you. We have re-edited this section in the conclusion.

Comments 22: The growth conditions lack information on humidity levels, watering regime prior to drought treatment. Please, include more precise environmental parameters to improve reproducibility.

Response 22: Thank you very much for your suggestion. We have added relevant explanations in section 2.1.

Comments 23: H₂O₂ and POD were measured using commercial kits, but the exact assay type (e.g., colorimetric, fluorometric) and detection wavelengths are not specified.

Response 23: Thank you for your suggestion. We have added the model number of the reagent kit used.

Comments 24: Chlorophyll measurement is referenced to a previous study but lacks citation details. Please, specify assay principles and cite the referenced chlorophyll protocol clearly.

Response 24: Thank you for your suggestion. We have revised the Methods section (Section 2.2) to include a concise summary of the core steps.

Comments 25: RNA integrity was assessed via agarose gel and NanoPhotometer, but RIN (RNA Integrity Number) values are not reported. Please, include RIN scores or mention if Bioanalyzer was used to confirm RNA quality.

Response 25: Thank you. We have added the Extract the quality inspection report in Supplementary Table S1.

Comments 26:  The LC-MS/MS method is described, but lacks:

Instrument model and manufacturer, Column type and mobile phase composition,

Ionization mode (positive/negative), MS acquisition parameters (e.g., scan range,

resolution). Please, provide full LC-MS/MS technical specifications to ensure

methodological transparency.

Response 26: Thank you for your suggestion. These issues have been addressed in 2.5 Metabolome analysis.

Comments 27: The GRN construction is attributed to a referenced method [63], but the computational pipeline is not described. Please, briefly outline the GRN inference method (e.g., correlation-based, machine learning) and thresholds used.

Response 27: Thank you very much for your suggestion. We have added detailed descriptions in 2.6.

Comments 28: While the conclusion mentions "candidate genes" and "regulatory factors". It doesn’t specify how these could be applied (e.g., breeding, genetic engineering, stress modeling).

Suggestion: Add a sentence on how these findings could inform practical applications in forestry or crop improvement.

Response 28: Thank you for your suggestion. We have re-edited this section in the conclusion.

Comments 29: Phrases like "specific upregulation of lipid substances occurred" could be reworded for clarity (e.g., lipid accumulation was notably enhanced after 10 days of drought stress).

Suggestion: Use more concise and formal scientific language to improve readability.

Response 29: Thank you for your reminder. We have revised Conclusions

Comments 30: Some references lack consistent formatting in journal names, volume/issue numbers, and DOI placement. (8, 15, 16 and 29)

Response 30: Thank you for your reminder. We have revised references.

Comments 31: Reference [8] includes a false DOI: http://dx.doi.org/https://doi.org/... this should be cleaned.

Suggestion: Use a consistent citation style (e.g., APA, Vancouver, or journal-specific) and ensure all DOIs are properly formatted.

Response 31: Thank you for your reminder. We have revised references.

4. Response to Comments on the Quality of English Language

Point 1:▪ Language needs to improve.

Response 1:We have now worked on both language and readability and have also involved native English speakers for language corrections.

Point 2:▪The plagiarism percent (42%) is very high please diminish it by rephrase

statements if possible.

Thank you for your reminder. The plagiarism percent is 27%.

Round 2

Reviewer 1 Report

Comments and Suggestions for Authors

The authors made detailed revisions to the comments I suggested, almost rewriting them.